# Sex-regulated gene dosage effect of PPARα on synaptic plasticity

Nathalie Pierrot[1,2] , Laurence Ris[3], Ilie-Cosmin Stancu[1,2,4] , Anna Doshina[1,2], Floriane Ribeiro[1,2], Donatienne Tyteca[1,5], Eric Baugé[6], Fanny Lalloyer[6], Liza Malong[1,2] , Olivier Schakman[1,2], Karelle Leroy[7], Pascal Kienlen-Campard[1,2] , Philippe Gailly[1,2] , Jean-Pierre Brion[7] , Ilse Dewachter[1,2,4], Bart Staels[6], Jean-Noël Octave[1,2]

**Mechanisms driving cognitive improvements following nuclear receptor activation are poorly understood. The peroxisome proliferator–activated nuclear receptor alpha (PPARα) forms heterodimers with the nuclear retinoid X receptor (RXR). We report that PPARα mediates the improvement of hippocampal synaptic plasticity upon RXR activation in a transgenic mouse model with cognitive deficits. This improvement results from an increase in GluA1 subunit expression of the alpha-amino-3-hydroxy-5-methyl-4-isoxazolepropionic acid (AMPA) receptor, eliciting an AMPA response at the excitatory synapses. Associated with a two times higher PPARα expression in males than in females, we show that male, but not female, PPARα null mutants display impaired hippocampal long-term potentiation. Moreover, PPARα knockdown in the hippocampus of cognition-impaired mice compromises the beneficial effects of RXR activation on synaptic plasticity only in males. Furthermore, selective PPARα activation with pemafibrate improves synaptic plasticity in male cognition-impaired mice, but not in females. We conclude that striking sex differences in hippocampal synaptic plasticity are observed in mice, related to differences in PPARα expression levels.**

## Introduction

The nuclear receptor (NR) superfamily of ligand-dependent transcription factors are broadly implicated in a wide variety of biological processes regulating energy balance, inflammation, lipid, and glucose metabolism (Evans & Mangelsdorf, 2014). NRs play an important role in the adaptive responses to environmental changes by controlling directly the expression of target genes through binding to sequence-specific elements located in gene regulatory regions (Evans & Mangelsdorf, 2014). Among NRs,

peroxisome proliferator–activated receptors (PPARs) and the liver X receptors (LXRs) form obligate heterodimers with retinoid X receptors (RXRs). PPAR/RXR and LXR/RXR heterodimers are permissive, meaning that receptor dimers can be activated by ligands for either partner in the dimer, or even by both synergistically (Evans & Mangelsdorf, 2014).

PPARs, including PPARα, PPARβ/δ, and PPARγ, are master metabolic regulators in response to dietary changes. PPARα plays an important role in the regulation of fatty acid (FA) catabolism (Staels et al, 1998). LXRs isoforms (LXRα and LXRβ) are involved in lipogenesis and reverse cholesterol transport (Bensinger & Tontonoz, 2008). Furthermore, PPARs and LXRs have also anti-inflammatory effects because they repress transcription of genes encoding pro-inflammatory cytokines (reviewed in Bensinger & Tontonoz (2008)).

These nuclear receptors are abundantly expressed in metabolically active tissues, including the brain of rodents and humans (Warden et al, 2016). Because of their anti-inflammatory and potential neuroprotective effects, PPARs, LXRs, and RXRs activation with specific agonists emerged as promising approaches for treating brain pathologies in several mouse models of Parkinson, Huntington, Alzheimer diseases, multiple and amyotrophic lateral sclerosis, stroke, and even in a mouse model with physiological brain aging–dependent cognitive decline (reviewed in Moutinho & Landreth (2017); Zolezzi et al (2017)).

Recent data indicate that activation of RXRs (Mariani et al, 2017) or PPARs (Roy et al, 2013) up-regulates the expression of a set of synaptic-related proteins involved in excitatory neurotransmission. Moreover, RXR activation increases dendritic complexity and branching of neurons promoting their differentiation and development (Mounier et al, 2015; Nam et al, 2016). However, the link between NRs activation and the improvement of synaptic plasticity is missing.

In the present work, we analyzed how RXR activation improves synaptic plasticity and neuronal function and identified PPARα as a crucial player. Upon RXR activation, the PPARα-dependent

[1]Université Catholique de Louvain, Brussels, Belgium  [2]Institute of Neuroscience, Brussels, Belgium  [3]Laboratory of Neuroscience, Health Institute, University of Mons, Mons, Belgium  [4]Biomedical Research Institute, Hasselt University, Hasselt, Belgium  [5]de Duve Institute, Brussels, Belgium  [6]Université de Lille EGID, Inserm, CHU Lille, Institut Pasteur de Lille, Lille, France  [7]Laboratory of Histology and Neuropathology, Université Libre de Bruxelles, Brussels, Belgium

Correspondence: nathalie.pierrot@uclouvain.be

up-regulation of GluA1 subunit-containing AMPA receptors mediates long-term potentiation (LTP) improvement in transgenic mice and AMPA responses in cortical cells. Associated with a higher expression of PPARα in males than in females, the absence of PPARα severely impairs LTP and GluA1 expression only in males. Knockdown of PPARα in the hippocampus of cognition-impaired mice abrogates the beneficial effects of RXR activation only in males. In these mice, treatment with pemafibrate, a highly potent selective PPARα activator (Yamazaki et al, 2007; Hennuyer et al, 2016), improves synaptic plasticity only in males, demonstrating a key role of PPARα in the regulation of synaptic function in a sex-specific manner.

# Results

### Synaptic plasticity, AMPA responses, and GluA1 expression are improved upon RXR activation

We first assessed in vivo the effect of RXR activation on synaptic plasticity in a well-characterized transgenic (Tg) mouse model of Alzheimer's disease (AD) (5xFAD), in which age-dependent synaptic and cognitive deficits occur (Oakley et al, 2006). We measured LTP in the hippocampal CA3-CA1 synapses, which are defined as an activity-dependent enhancement of synaptic strength involved in memory processing (Bliss & Collingridge, 1993). Impaired LTP found in Tg 5xFAD hippocampus was recovered ($P < 0.0001$) after oral administration of bexarotene for 12 d and became similar to vehicle-treated control mice (Fig 1A). Bexarotene did not improve LTP of Wt mice (Fig S1A). The efficiency of the treatment of Tg mice could result from a breakdown of the blood–brain barrier in 5XFAD mice (Montagne et al, 2017). This recovery of LTP in 5xFAD mice was observed together with improved cognition in the object recognition and spatial navigation tasks, which was independent of amyloid plaque load in different regions of the brain (Fig S1B–E).

We next analyzed whether the RXR activation–mediated improvement of LTP was related to changes in expression levels of N-methyl-D-aspartate receptors (NMDARs) and AMPA receptors (AMPARs), known to be required for LTP at the excitatory synapses (Bliss & Collingridge, 1993). We, therefore, measured the expression of both GluN2A-, GluN2B-containing NMDARs, and GluA1-containing AMPARs in hippocampal lysates of these mice. Whereas GluN2A ($P = 0.0070$), 2B ($P = 0.0019$), and GluA1 ($P = 0.0007$) decreased in Tg mice compared with Wt mice, a 12-d treatment of Tg mice with the RXR agonist bexarotene specifically increased GluA1-containing AMPARs ($P = 0.0379$) (Fig 1B). These results indicate that improvement in synaptic plasticity by RXR activation is tightly associated with an increased expression of GluA1 subunit in treated Tg animals.

The RXR activation–mediated GluA1 increase described above could have an impact on basal glutamatergic responses. Rat cortical cells in culture were treated or not with bexarotene (100 nM) for 24 h, and NMDARs and AMPARs subunits measured. At 13–14 d in vitro (DIV), GluN2A ($P = 0.0655$) and 2B ($P = 0.2916$) were unchanged by bexarotene treatment (Fig 1C). In contrast, GluA1 protein was increased ($P = 0.0003$) in these cells treated with bexarotene (Fig 1C). Increase in GluA1 protein level was also observed in hippocampal neurons ($P = 0.0133$) and in 7 DIV ($P = 0.0273$)–cultured hippocampal slices incubated with a higher bexarotene concentration of 300 nM (Figs S2A and 2B).

Because activation of AMPA and NMDA receptors mediates $Ca^{2+}$ entry into cells, we monitored AMPA and NMDA-induced $Ca^{2+}$ responses in cortical cell cultures with the Fura-2 AM $Ca^{2+}$-sensitive dye by using single-cell calcium imaging. Measurements of fluorescence intensity changes showed that only AMPA ($P < 0.0001$) (but not NMDA, $P > 0.9999$) elicited a stronger $Ca^{2+}$ increase with a larger amplitude in bexarotene-treated than in control cortical cells (Fig 1D). The higher $Ca^{2+}$ permeability of GluA1-containing AMPARs observed in bexarotene-treated cortical cells did not result from changes in GluA2 expression, a subunit known to modify AMPARs properties by forming heteromeric complexes with GluA1 (reviewed in Derkach et al (2007)) (Fig S2C).

To address whether RXR activation induces membrane insertion of GluA1-containing AMPARs, we quantified GluA1 expressed at the cell surface following biotinylation of cell surface proteins in cortical cells treated or not with bexarotene. Activation of RXR increased GluA1 protein levels in both the total ($P = 0.0002$) and biotinylated ($P = 0.0273$) fraction as compared with control (Fig 1E). We next tested the influence of bexarotene treatment on the synaptic localization of GluA1-containing AMPARs, by measuring their co-localization with SynGAP, a Ras-GTPase–activating protein highly enriched at excitatory synapses (Chen et al, 1998). GluA1 fluorescence intensity was higher and exhibited a more punctuated pattern in bexarotene-treated cells compared with control (Fig 1F and G). When postsynaptic puncta were quantified, GluA1-containing AMPARs were increased by bexarotene compared with control (Fig 1H) and exhibited a stronger overlap with the SynGAP postsynaptic marker after bexarotene treatment (Fig 1H). Concomitantly, a decrease in the number of SynGAP peaks by bexarotene was observed compared with control (Fig 1H). In addition, a significant twofold increase in the average cluster size of GluA1 puncta ($P < 0.0001$) was observed when comparing treated with control cells (Fig 1I). Together, these results support the hypothesis that RXR activation improves AMPA responses by increasing GluA1 expression and its targeting to the excitatory synapses.

### PPARα is necessary for RXR activation–mediated improvements

Next, we investigated the cellular mechanisms by which RXR activation increases the expression of the GluA1-containing AMPARs. We first analyzed whether the expression of the cAMP response element binding (CREB) protein, involved in the synaptic maintenance of GluA1 subunit (Middei et al, 2013), was responsive to RXR activation (Nam et al, 2016). Both GluA1 ($P = 0.0486$) and CREB mRNA ($P = 0.0007$) as well as CREB protein ($P = 0.0006$) levels and immunostaining intensity were increased in bexarotene-treated cortical cells compared with control (Figs S3A and 3B).

Because RXR forms dimeric complexes with other NRs and that autoregulation and cross-regulation of NRs have been described (Tata, 1994; Lefebvre et al, 2010), we wondered whether the bexarotene-mediated RXR activation could modulate expression

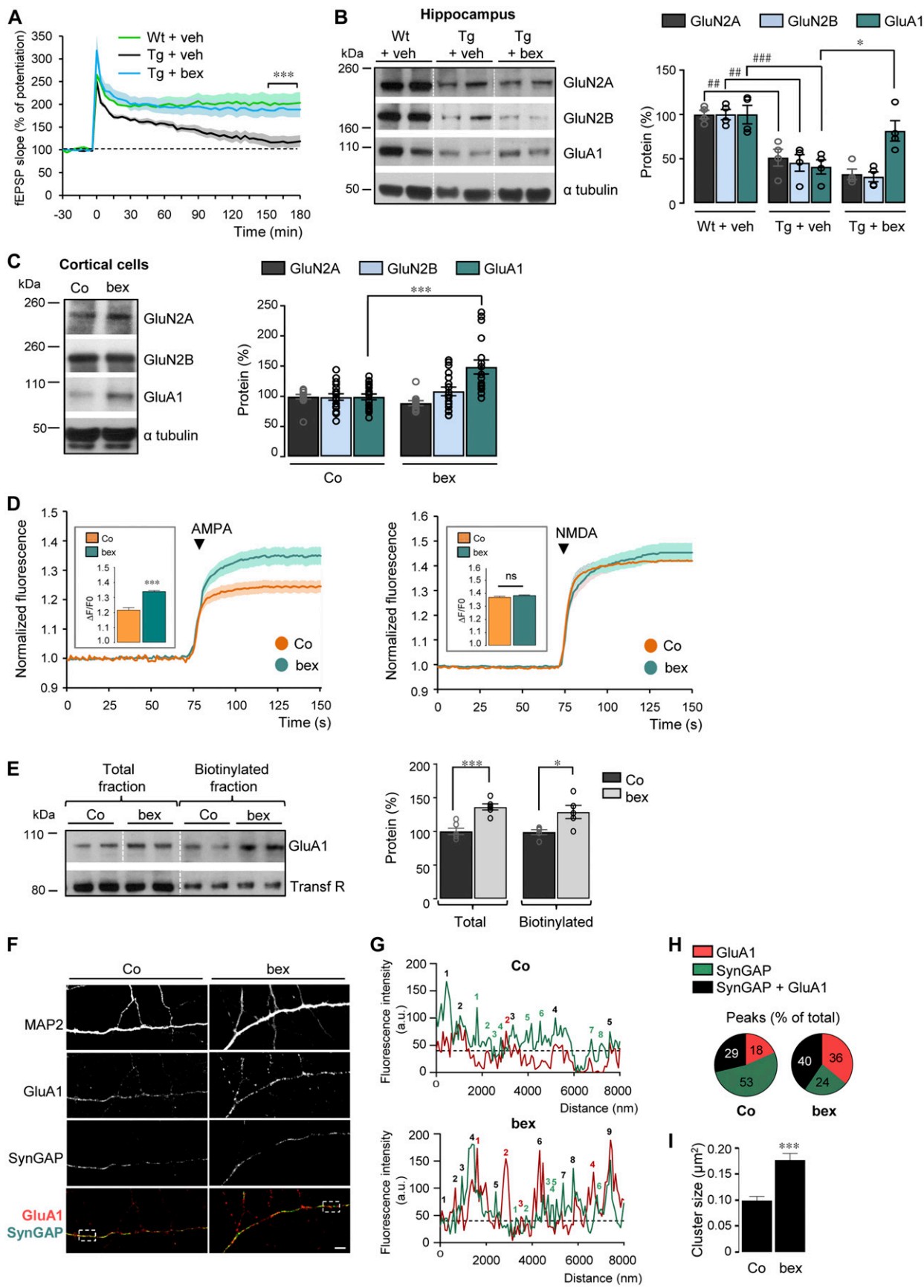

levels of the most prominent NRs found as obligate permissive heterodimers with RXR in neuronal and non-neuronal cells (reviewed in Bookout et al (2006); Zolezzi et al (2017)). We did not observe any modification in mRNA levels of RXRα (*P* > 0.9999), β (*P* = 0.8478), γ (*P* > 0.9999), PPARβ (*P* = 0.2070), PPARγ (*P* = 0.3110), and LXRβ (*P* = 0.6481) isoforms in cortical cells treated with bexarotene (Fig 2A). In contrast, we observed a twofold increase in PPARα (*P* = 0.0005) mRNA levels (Fig 2A) together with an increased immunostaining of PPARα in bexarotene-treated cortical cells compared with control (Fig S3C). These results indicate that bexarotene-mediated RXR activation specifically increases PPARα expression.

We next analyzed whether up-regulation of GluA1 expression by RXR activation depends on PPARα expression. We first measured GluA1 expression in cultured cortical cells from wild-type (*Wt*) and PPARα-deficient (*Ppara⁻/⁻*) mice (Figs S4A and 4B). As expected (Roy et al, 2013), absence of PPARα decreased GluA1 expression at mRNA (*P* = 0.0079) and protein (*P* < 0.0001) levels in cortical cells (Figs S4C and 4D). In addition, the lower GluA1 expression in *Ppara⁻/⁻* cells was consistent with a decreased AMPA-induced Ca²⁺ response (*P* < 0.0001) in these cells (Fig S4E). A PPAR-responsive element was recently identified in the *Creb* promoter, identifying it as a PPARα target (Roy et al, 2013), and we show that CREB mRNA (*P* = 0.0079) levels are decreased in *Ppara⁻/⁻* cells (Fig S4C).

To address whether the RXR activation-mediated GluA1 up-regulation could be PPARα dependent, *Wt* and *Ppara⁻/⁻* cultured cortical cells were treated or not with bexarotene. PPARα deficiency totally prevented the increase in GluA1 mRNA and protein levels (*P* = 0.6385 and *P* = 0.1801) observed in *Wt* cells treated with bexarotene (Fig 2B–D). On the contrary, the expression of ATP-binding cassette transporter A1 (ABCA1), a membrane protein driving cholesterol efflux (Venkateswaran et al, 2000), known to be regulated by LXR upon RXR activation, was still increased by bexarotene in *Ppara⁻/⁻* cortical cells (*P* < 0.0001, Fig 2B and *P* = 0.0023, Fig 2D). This clearly demonstrates that GluA1 but not ABCA1 expression is regulated by the RXR/PPARα heterodimer.

Measurements of fluorescence intensity changes showed that AMPA elicited a greater Ca²⁺ increase with a larger amplitude only in *Wt* (*P* < 0.0001) but not in *Ppara⁻/⁻* (*P* > 0.9999) cells treated with bexarotene (Fig 2E and F). Thus, increased expression of the GluA1 subunit of AMPARs by activation of RXR is PPARα dependent.

## PPARα deficiency impairs LTP and GluA1 expression in male mice

PPARα is required for normal cognitive function (D'Agostino et al, 2015; Roy et al, 2013). As previously reported (Dotson et al, 2016), PPARα mRNA levels (*P* < 0.0001) are higher in the hippocampus of male than female mice (Fig 3A) prompting us to study potential sex different responses. Surprisingly, LTP induced by a single tetanus was significantly larger (*P* < 0.0001) in 5–6-mo-old males than in females (Fig 3B). Interestingly, GluA1 mRNA (*P* = 0.0021) and protein (*P* = 0.0058) expression levels were higher in *Wt* male than female mice (Fig 3C and D), although similar GluN2A and 2B mRNA and protein levels were measured in males and females (Fig 3C and D).

Absence of PPARα expression in 5–6-mo-old *Ppara⁻/⁻* female mice did not influence either the potentiation induced by one train of stimulation or its maintenance as compared with *Wt* mice (Fig 3B and E). In contrast, both induction and maintenance of LTP were strongly reduced in *Ppara⁻/⁻* male mice compared with *Wt* mice (Fig 3B and E). A specific decreased expression of GluA1 was measured at the mRNA (*P* = 0.0012) and protein (*P* = 0.0003) level in *Ppara⁻/⁻* male mice but not in females (Fig 3F and G). Taken together, these results suggest that PPARα induces sex-dependent modifications in LTP by specifically affecting the expression of the GluA1 subunit of AMPARs only in male mice.

## Synaptic plasticity improved by RXR activation is PPARα and sex dependent

We next wondered whether the improved synaptic plasticity and GluA1 expression observed in bexarotene-treated Tg animals (5xFAD mice) are mediated by PPARα. Because disruption of PPARα decreases lifespan in 5xFAD mice (Corbett et al, 2015), we decided to acutely decrease PPARα expression in the hippocampus of 9–10-mo-old Tg mice by using a serotype 9 adeno-associated viruses (AAVs) coding an shRNA construct designed to target endogenous PPARα (AAV-ShPpara). A scrambled ShRNA (AAV-ShSc) was used as control. We first tested the efficiency of AAV-ShPpara construct in vitro following transduction of cultured cortical cells at 4 DIV. 10 d after transduction, PPARα immunoreactivity was significantly decreased (*P* = 0.0152) in AAV-ShPpara compared with AAV-ShSc–transduced cells (Fig S5A). Neuronal activity measured by spontaneous calcium oscillations and amplitude of AMPA-induced Ca²⁺ responses were reduced (*P* < 0.0001) in AAV-ShPpara compared

**Figure 1. RXR activation ameliorates LTP, AMPA-induced responses, and GluA1 expression.**
**(A, B)** Transgenic (Tg) 5xFAD and wild-type (Wt) mice treated at 9–10 mo with bexarotene (bex) (100 mg/kg/d) or vehicle (veh) by gavage (12 d). **(A)** CA1 LTP in hippocampal slices of Tg + veh (n = 8) and Tg + bex (n = 7) compared with Wt + veh (n = 6). \*\*\**P* < 0.001, *t* test. **(B)** Representative Western blot of hippocampal lysates from Wt and Tg mice. Right panel: GluN2A, GluN2B, and GluA1/α tubulin ratios. Results are expressed as percentage of Wt + veh (n = 5 for each); compared with Wt + veh: ##*P* < 0.01, ###*P* < 0.001 compared with Tg + veh: \**P* < 0.05, ANOVA followed by Bonferroni's multiple-comparison posttest. **(C)** Representative Western blot of control (Co) and 100 nM bex-treated (24 h) cell lysates from cortical cultures. Right panel: GluN2A, 2B, and GluA1/α tubulin ratios. Results are expressed as percentage of respective Co (Co; n = 18 and bex; n = 17 of each analyzed in 10 independent experiments, \*\*\**P* < 0.001, Mann–Whitney test). **(D)** Normalized fluorescence intensity of Fura-2 AM in Co and bex-treated cortical cells in the presence of AMPA (right panel) and NMDA (left panel). Insets: amplitude of AMPA (Co; n = 257 and bex; n = 148 cells analyzed in six and five independent experiments) and NMDA responses (Co; n = 307 and bex; n = 177 in seven and six independent experiments); \*\*\**P* < 0.001, ns: not significant (*P* > 0.05), Mann–Whitney test). **(E)** Cell surface biotinylation analyzed by Western blot (left panel). Right panel: GluA1/transferrin receptor (Transf R) ratios. Results are expressed as percentage of respective Co (n = 6 of each in three independent experiments, \**P* < 0.05, \*\*\**P* < 0.001, *t* test). **(F)** Co and bex-treated MAP2-positive neurons immunolabelled for GluA1 (red) and SynGAP (green). Scale bar: 5 μm. **(F, G, H)** Representative profiles of the regions highlighted by the rectangles in merge pictures in (F). In (H), the number of GluA1 (red numbers), SynGAP (green numbers), and GluA1/SynGAP overlapping (black numbers) peaks were quantified on >60 profiles per condition from four independent experiments. Results are expressed as percentage of total peaks from all the profiles. **(I)** Quantification of synaptic GluA1 puncta-cluster size (in μm²) in Co or bex-treated neurons. Results of >900 (from 23 images) and 1700 clusters (from 33 images) for Co and bex-treated neurons, respectively. \*\*\**P* < 0.001, *t* test. Data information: data are presented as mean ± SEM. Source data are available for this figure.

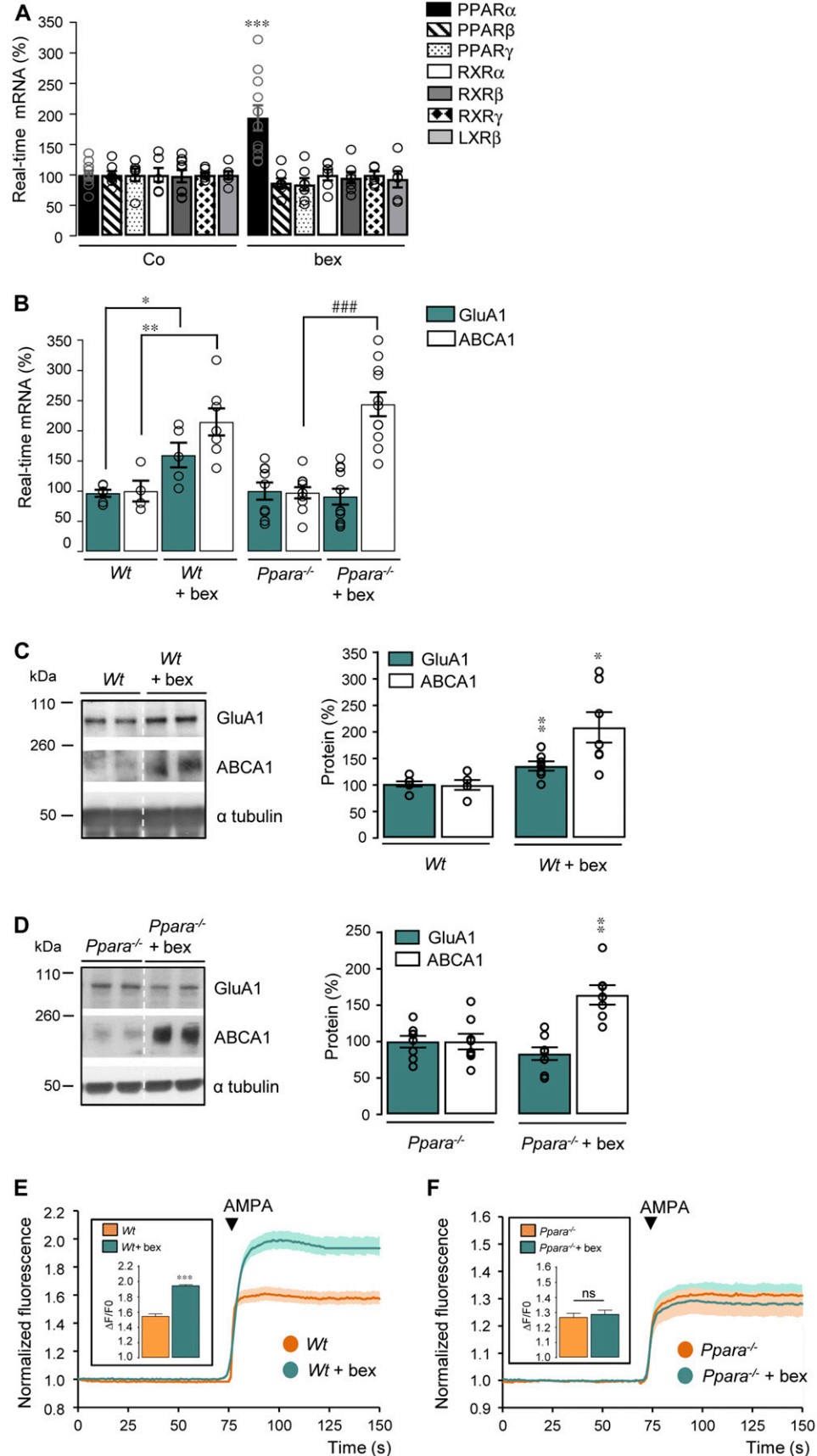

**Figure 2. Absence of PPARα abrogates GluA1 expression and AMPA responses induced by RXR activation.**
**(A)** RT-qPCR analyses of *Ppara*, *Pparb*, *Pparg*, *Rxra*, *Rxrg*, and *Nr1h2* mRNA levels in control (Co) and bexarotene (bex, 100 nM/24 h)-treated cortical cultures (n = 6–10 in 3–5 independent experiments), ***$P < 0.001$, $t$ test. **(B)** RT-qPCR analyses of *Gria1* and *Abca1* mRNA levels in three independent experiments from cortical cells prepared from wild-type (*Wt*; n = 6 of each) and *Ppara*-deficient (*Ppara*[−/−]; n = 10 of each) mice treated or not with bex. Results are expressed as percentage of corresponding non-treated cells (compared with *Wt*: *$P < 0.05$, **$P < 0.01$; compared with *Ppara*[−/−]: ###$P < 0.001$ [$t$ test]). **(C, D)** Representative Western blots of cortical cell lysates from *Wt* (C) and *Ppara*[−/−] (D) cultures treated or not with bex. Right panels: GluA1 and ABCA1/α tubulin ratios. Results are expressed as percentage of respective untreated *Wt* or *Ppara*[−/−] (*Wt*; n = 6, *Wt* + bex; n = 7, *Ppara*[−/−]; n = 8, and *Ppara*[−/−] + bex; n = 7 of each analyzed in three independent experiments; *$P < 0.05$, **$P < 0.01$, $t$ test, except for ABCA1 in *Wt* + bex: Mann–Whitney test). **(E, F)** AMPA-induced calcium fluorescence in *Wt* (E) and *Ppara*[−/−] (F) cortical cells treated with bex. Insets: AMPA responses amplitude (*Wt*; n = 320, *Wt* + bex; n = 118, *Ppara*[−/−]; n = 430, and *Ppara*[−/−] + bex; n = 374 cells analyzed in three to six independent experiments; ***$P < 0.001$, ns: not significant [$P > 0.05$], Mann–Whitney test). Data information: data are presented as mean ± SEM. Source data are available for this figure.

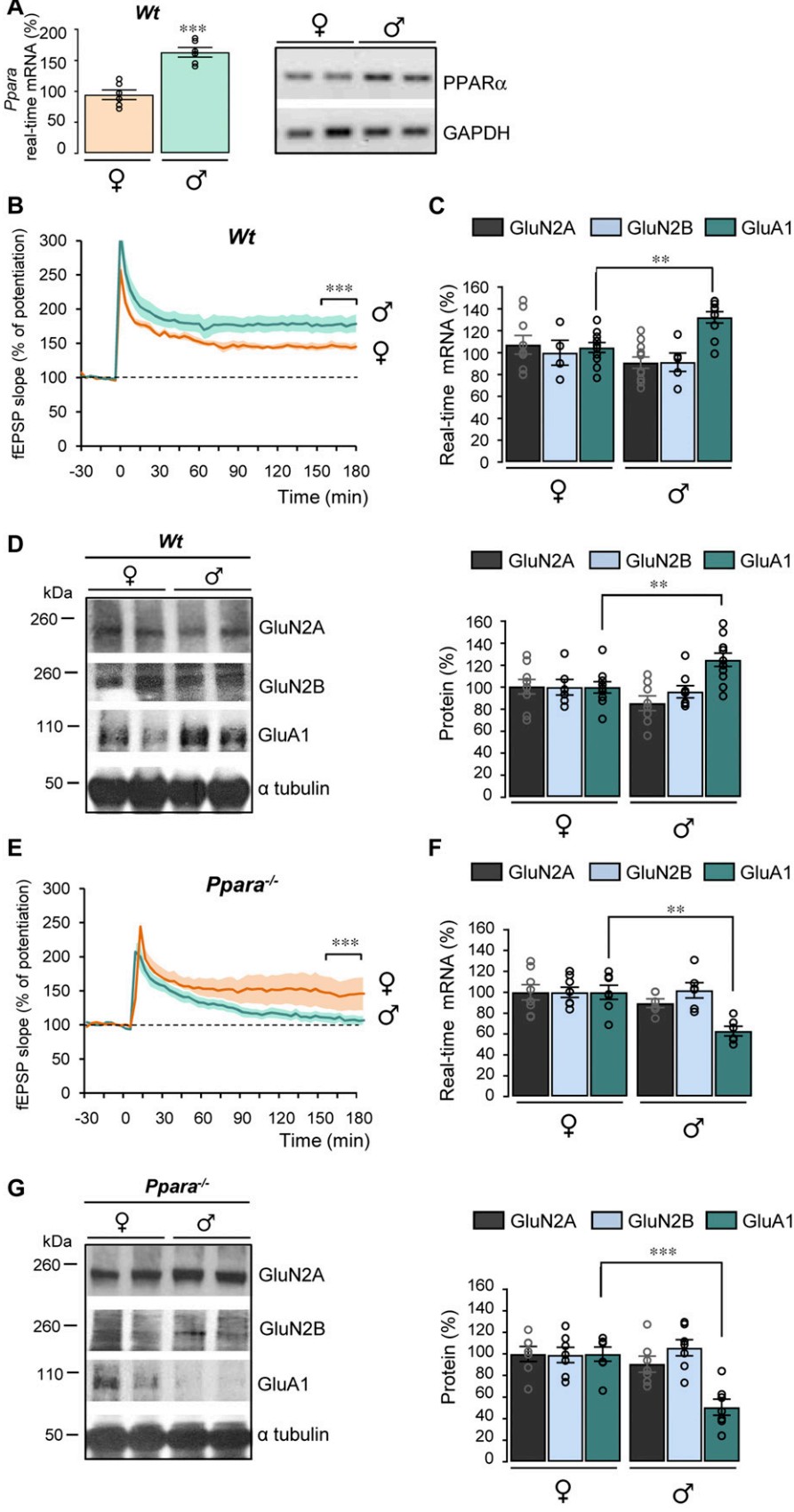

**Figure 3. PPARα deficiency impairs LTP and GluA1 expression in male mice.**

**(A)** PPARα mRNA levels analyzed in the hippocampus from female (♀) and male (♂) wild-type (*Wt*) mice by RT-qPCR and semi-quantitative RT-PCR (left and right panels, respectively). Results are expressed as percentage of corresponding ♀ (n = 6 of each, ***P < 0.001, *t* test). **(B, E)** CA1 LTP in hippocampal slices from 5–6-mo-old male (♂) and female (♀) wild-type (*Wt*, in (B)) and *Ppara*-deficient (*Ppara*⁻/⁻, in (E)) mice (n = 6 in each group). ***P < 0.001, *t* test. **(C, F)** RT-qPCR analyses of *Grin2A* and *2B* and *Gria1* mRNA levels in the hippocampus from female (♀) and male (♂) *Wt* (n = 11) and *Ppara*⁻/⁻ (n = 6–8) mice. Results are expressed as percentage of corresponding ♀ (in (C) compared with *Wt* ♀: **P < 0.01, Mann–Whitney test; in (F) compared with *Ppara*⁻/⁻ ♀: **P < 0.01, *t* test). **(D, G)** Representative Western blots of hippocampal lysates from female (♀) and male (♂) *Wt* and *Ppara*⁻/⁻ mice. Right panels: quantification of GluN2A, GluN2B, and GluA1/α tubulin ratios. Results are expressed as percentage of corresponding ♀ (in (D) compared with *Wt* ♀: **P < 0.01, in (G) compared with *Ppara*⁻/⁻ ♀: ***P < 0.001, *t* test) (*Wt*; n = 8–10 and *Ppara*⁻/⁻; n = 7 for each condition). Data information: data are presented as mean ± SEM. Source data are available for this figure.

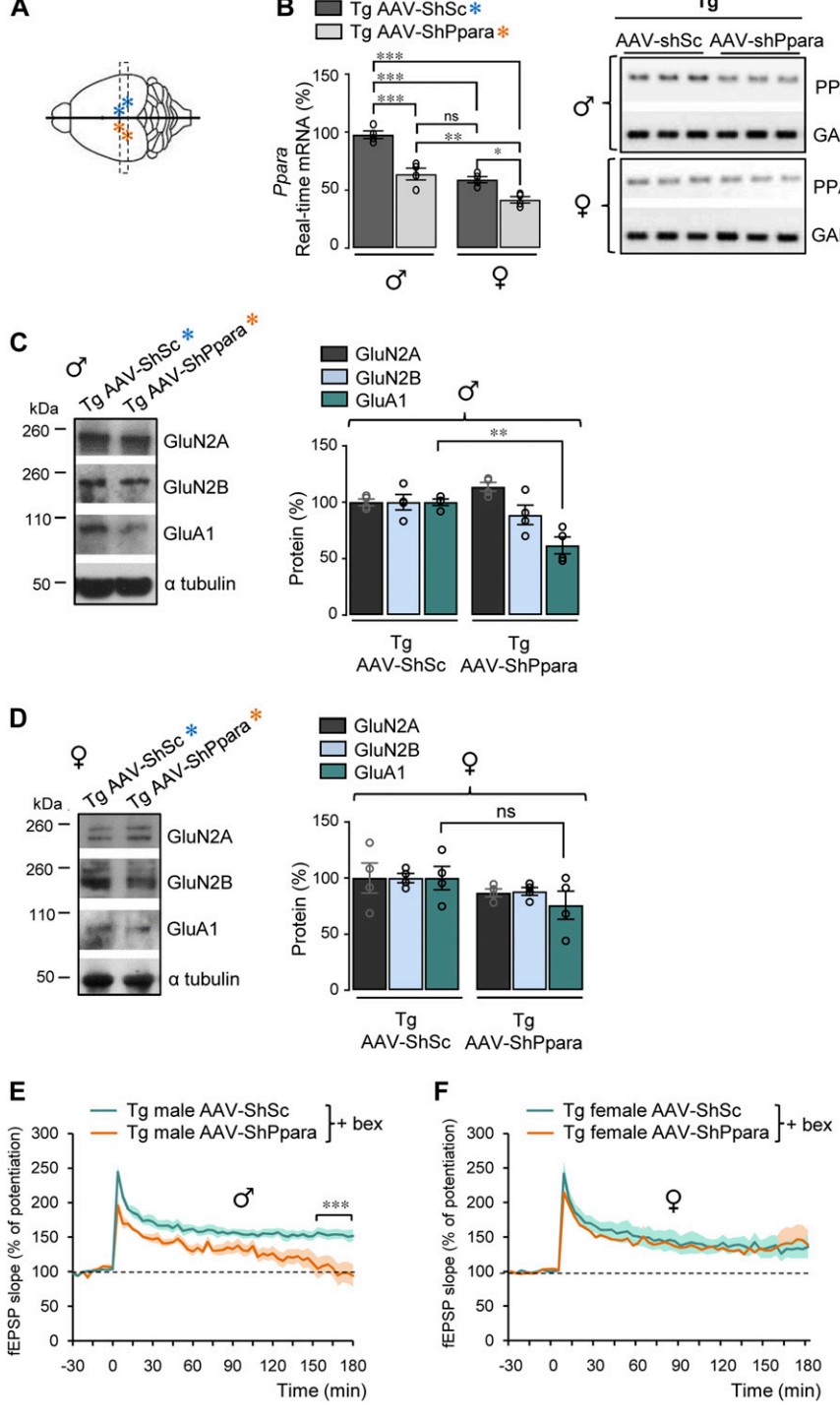

**Figure 4. LTP improvement by RXR activation is PPARα and sex dependent.**
**(A)** Schematic drawing of the top view of a mouse brain. Stars: ipsi and contralateral injection sites of *Ppara* and scramble shRNA AAV (AAV-ShPpara [orange] and AAV-ShSc [blue]). **(B–D)** Dashed line represents the plane of the coronal section used in (B–D) for biochemical analyses. **(B)** RT-qPCR analyses of *Ppara* mRNA levels in male (♂) and female (♀) Tg mice hippocampi AAV-ShSc and AAV-ShPpara injected at 9–10 mo. Results are expressed as percentage of AAV-ShSc–injected male mice (n = 4 of each; *P < 0.05, **P < 0.01, ***P < 0.001, ns: not significant (P > 0.05), ANOVA followed by Bonferroni's multiple-comparison posttest). Right panels: PPARα semi-quantitative RT-PCR. **(C, D)** Representative Western blots of hippocampal lysates from male (♂, in C) and female (♀, in D) Tg mice AAV-ShSc and AAV-ShPpara injected. Right panels: quantification of GluN2A, GluN2B, and GluA1/α tubulin ratios in male (in C) and female (in D) Tg mice AAV-ShSc and AAV-ShPpara injected. Results are expressed as percentage of corresponding Tg mice AAV-ShSc injected (n = 4 in each condition, **P < 0.01, t test; ns: not significant [P > 0.05]). **(E, F)** CA1 LTP in hippocampal slices from male (♂, in (E)) and female (♀, in (F)) transgenic (Tg) 5xFAD mice (9–10 mo) AAV-ShPpara and AAV-ShSc injected and perfused with 4 µM bexarotene (bex) (n = 4 in each group, ***P < 0.001, t test). Data information: data are presented as mean ± SEM.
Source data are available for this figure.

with AAV-ShSc infected cells (Figs S5B and 5C, respectively). PPARα knockdown, although not affecting ABCA1 expression (P > 0.9999), decreased GluA1 mRNA (P = 0.0005) and protein (P < 0.0001) levels by about 50% (Fig S5D). In addition, PPARα knockdown abolished the increase in GluA1 mRNA and protein levels (P = 0.6051 and P = 0.1655, respectively) observed in AAV-ShSc cells treated with bexarotene (100 nM, 24 h) (Figs S6A and 6B). On the contrary, ABCA1 mRNA and protein expression were still induced in AAV-ShSc (P = 0.0058 and P < 0.0001, respectively) and AAV-ShPpara (P = 0.0052 and P < 0.0001, respectively) cortical cells treated with bexarotene (Figs S6A and 6B). Measurement of fluorescence intensity changes showed that AMPA elicited a greater $Ca^{2+}$ increase (P < 0.0001) with a larger amplitude only in AAV-ShSc but not in AAV-ShPpara (P > 0.9999)–transduced cells treated with bexarotene (Figs S6C and 6D).

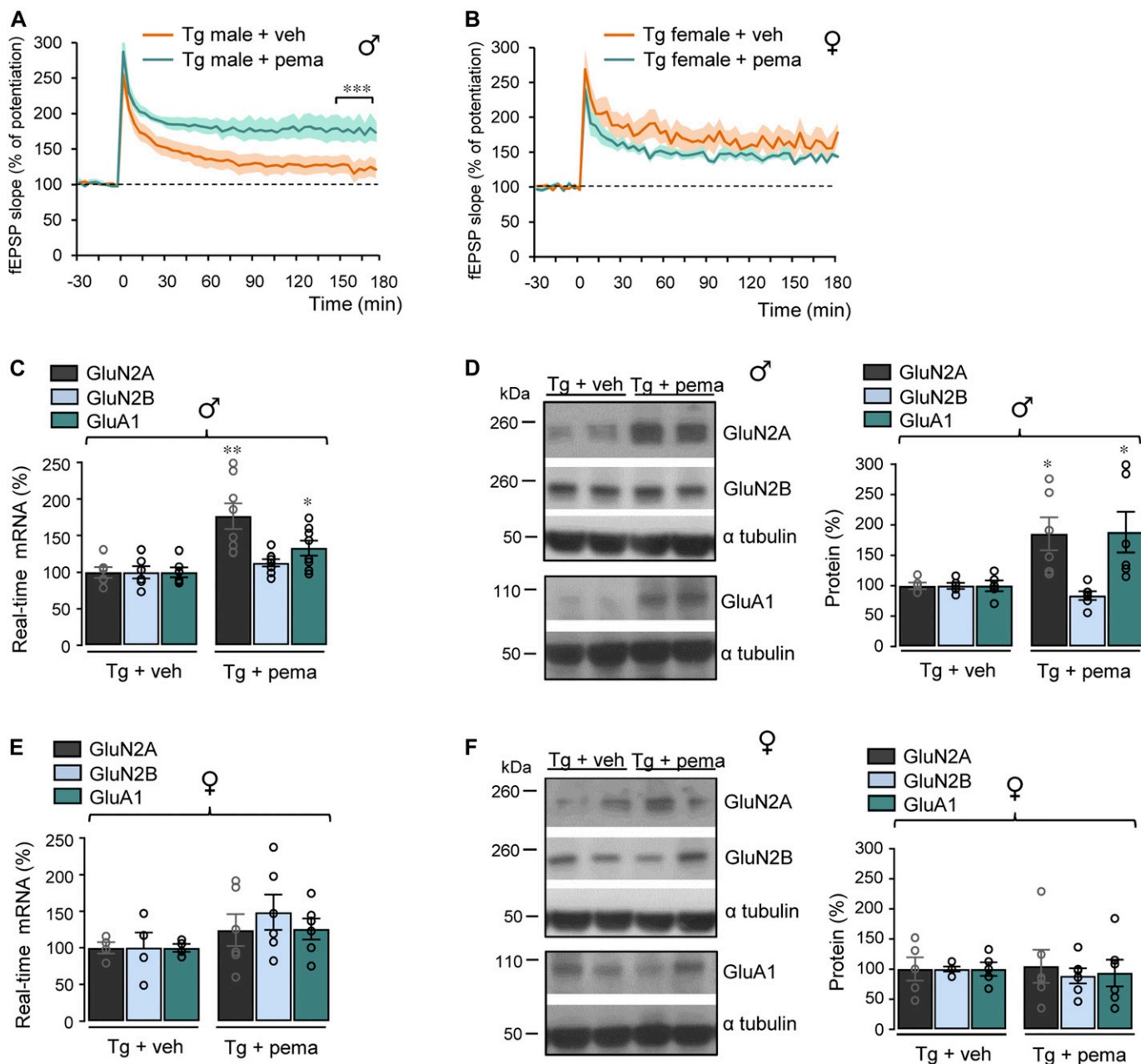

**Figure 5.    Sex-specific improvement of LTP by pemafibrate.**
**(A, B)** Male (♂) and female (♀) transgenic (Tg) 5xFAD mice treated at 12 mo with pemafibrate (pema) (1 mg/kg/d) or vehicle (veh) by gavage (12 d). CA1 LTP in hippocampal slices of male Tg + veh (n = 6) and Tg + pema mice (n = 8) in (A) and female Tg + veh (n = 4) and Tg + pema mice (n = 6) in (B), ***$P < 0.001$, $t$ test. **(C, E)** RT-qPCR analyses of *Grin2A* and *2B* and *Gria1* mRNA levels in the hippocampus from male Tg mice treated with pema (n = 8) or veh (n = 6 for each condition) in (C) and from female Tg mice treated with pema (n = 6) or veh (n = 4 of each) in (E). Results are expressed as percentage of corresponding Tg + veh (*$P < 0.05$, **$P < 0.01$; $t$ test). **(D, F)** Representative Western blots of hippocampal lysates from male (D) and female (F) Tg mice treated or not with pema. Right panels: quantification of GluN2A, GluN2B, and GluA1/α tubulin ratios. Results are expressed as percentage of corresponding Tg + veh (Tg + veh n = 5 and Tg + pema n = 6 of each, *$P < 0.05$, $t$ test). Data information: data are presented as mean ± SEM.
Source data are available for this figure.

We next analyzed in vivo the effect of the acute knockdown of PPARα expression after stereotaxic injection of AAV-ShPpara and AAV-ShSc constructs in the right and left hippocampi of Tg mice, respectively (Fig 4A). Because PPARα and GluA1 expressions differ between males and females, the effect of knockdown of PPARα was studied in males and females separately. PPARα ($P = 0.0019$) and GluA1 ($P = 0.0103$) mRNA as well as GluA1 ($P = 0.0021$) protein levels were higher in males compared with females (Figs S7A and 7B). 3 wk after stereotaxic AAV injection, PPARα mRNA levels ($P = 0.0001$) significantly decreased in the hippocampi of male Tg mice injected

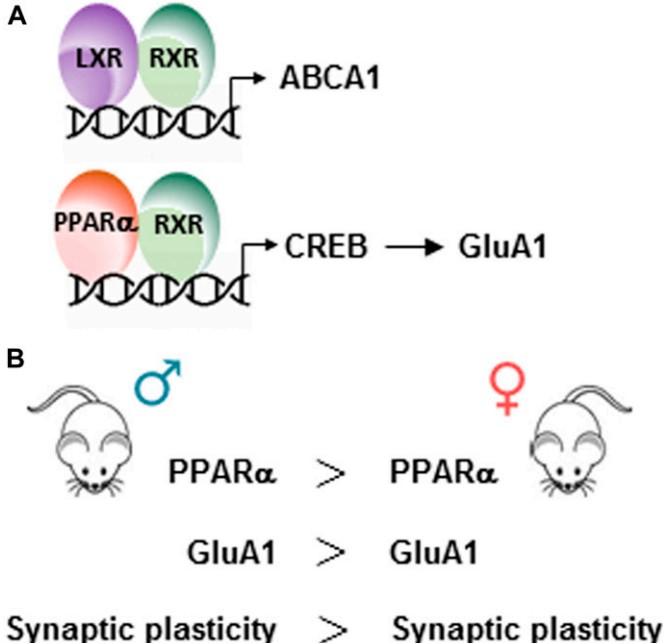

**Figure 6. Sex- and PPARα-specific effects on GluA1 expression and synaptic plasticity.**
**(A)** Schematic representation of LXR/RXR and PPARα/RXR heterodimers bound to a specific DNA sequence. LXR/RXR drives the expression of ABCA1 and PPARα/RXR drives the expression of CREB, which in turn, drives the expression of GluA1. **(B)** PPARα and GluA1 expression are higher (>) in male (♂) than in female (♀) mice. The gene dosage effect of PPARα on GluA1 expression leads to a better (>) synaptic plasticity in male than in female mice.

with AAV-ShPpara to levels similar ($P = 0.7796$) to those detected in female Tg mice injected with AAV-ShSc (Fig 4B). A significant, but less pronounced, decrease in PPARα mRNA ($P = 0.0239$) was also observed in female Tg mice injected with AAV-ShPpara (Fig 4B), an effect likely due to the lower baseline PPARα expression in female mice (Fig 4B). In association with this difference in PPARα expression between males and females, PPARα knockdown decreased GluA1 protein levels ($P = 0.0028$) only in hippocampal lysates from male Tg mice injected with AAV-ShPpara but not in females ($P = 0.1894$) (Fig 4C and D). GluN2A and 2B protein expression was not affected by the treatment (Fig 4C and D). Next, we wondered whether PPARα was needed for the recovery of synaptic plasticity induced by RXR activation. We first measured LTP on hippocampal slices from male Tg mice treated or not for 2.5 h with bexarotene 4 μM (Fig S7C). The results were similar to those obtained following gavage of Tg mice. LTP was then measured on hippocampal slices from male and female Tg mice injected with AAV-ShPpara or AAV-ShSc and incubated with bexarotene 4 μM. After 2.5 h of bexarotene incubation, the potentiation induced by high-frequency stimulation was lower in male Tg male mice injected with AAV-ShPpara compared with those injected with AAV-ShSc (Fig 4E). Moreover, LTP progressively decreased ($P < 0.0001$) in male Tg mice injected with AAV-ShPpara compared with male Tg mice injected with AAV-ShSc 3 h after the train of stimulation (Fig 4E). Both induction and maintenance of LTP were not affected in female Tg mice injected with AAV-ShPpara compared with AAV-ShSc

(Fig 4F). Taken together, these results show that improvement of synaptic plasticity by RXR activation is PPARα- and sex dependent.

## Sex-specific improvement of LTP by pemafibrate

We then tested the effect of direct activation of PPARα using pemafibrate, a selective agonist for PPARα (Yamazaki et al, 2007; Hennuyer et al, 2016). We first assessed in vivo the effectiveness of PPARα activation in 12-mo-old male and female Tg (5xFAD) mice treated for 12 d by oral gavage with pemafibrate (Hennuyer et al, 2016) or vehicle. As previously described in human hepatocytes and mouse liver (Raza-Iqbal et al, 2015), we observed an increase in mRNA levels of PDK4 ($P = 0.0049$), a PPARα target gene, in hippocampal lysates from Tg mice treated with pemafibrate (Fig S8A). These results demonstrate that oral administration of pemafibrate is able to activate PPARα in the brain.

We then measured the effect of pemafibrate on LTP. Although PPARα ($P = 0.0070$), PPARβ ($P = 0.0348$), and PPARγ ($P = 0.0083$) mRNA levels were lower in Tg compared with *Wt* mice (Fig S8B), the decreased LTP observed in male Tg mice treated with vehicle was nevertheless recovered ($P < 0.0001$) following treatment with pemafibrate (Fig 5A). Both induction and maintenance of LTP were strongly improved in male Tg mice with pemafibrate compared with vehicle (Fig 5A). However, activation of PPARα in female Tg mice did not influence LTP potentiation nor its maintenance compared with vehicle (Fig 5B). In male Tg mice treated with pemafibrate, mRNA and protein levels of the GluN2A ($P = 0.0037$ and $P = 0.0214$, respectively) and GluA1 ($P = 0.0276$ and $P = 0.0453$) subunits were significantly up-regulated compared with male Tg mice treated with vehicle (Fig 5C and D, respectively). No significant increase in the NMDARs and AMPARs subunits was measured in pemafibrate compared with vehicle-treated female Tg mice (Fig 5E and F). Taken together, these results show that activation of PPARα with pemafibrate improves synaptic plasticity in a sex-specific manner with a pharmacological response (increase in both GluN2A and GluA1 expression) different from that resulting from RXR activation (increase in GluA1 expression).

## Discussion

We report here that PPARα, a master metabolic regulator involved in FA catabolism (Staels et al, 1998), plays a central role in hippocampal synaptic plasticity by driving the expression of the GluA1 subunit of AMPARs in a sex-specific manner.

We show that LTP improvement observed in a Tg mouse model of AD upon RXR activation with bexarotene is concomitant with the specific up-regulation of GluA1 expression. Adult GluA1 knockout mice cannot generate LTP and have cognitive abnormalities (Schmitt et al, 2005), indicating that the GluA1 subunit plays a critical role in synaptic plasticity and cognition. Even in the absence of any modification in NMDARs subunits, we cannot rule out that changes in subunit composition or posttranslational modifications that affect NMDARs gating and trafficking and also AMPARs function (for review see Derkach et al (2007)), could occur following RXR activation.

In cortical cells in culture, we show that RXR activation with bexarotene induces a cell-autonomous increase in the expression of the GluA1 subunit. We show that GluA1 up-regulation induced by RXR activation was totally abolished in the absence of PPARα, whereas the expression of the LXR target gene, *ABCA1*, was still induced. Therefore, RXR/LXR activation increases *ABCA1* gene transcription, whereas RXR/PPARα activation drives the up-regulation of GluA1 expression (Fig 6A).

Because of differences in expression levels of PPARα between males and females (Dotson et al, 2016), a sex difference in the regulation of GluA1 expression and synaptic plasticity by PPARα was found (Fig 6B). Many different pathways, which do not involve PPARα, can control GluA1 expression, and we cannot conclude that LTP difference between WT males and females relies only on a differential expression of PPARα. Nevertheless, a two times higher expression of PPARα in males than in females induces a PPARα-mediated regulation of GluA1 expression only in males. This suggests that a threshold level of PPARα expression is needed to regulate GluA1 expression, and this level is not reached in females, which are insensitive to bexarotene treatment. Hormones are known to influence the expression of PPARα in a sex-specific manner because gonadectomy of male rats decreases PPARα expression levels (Jalouli et al, 2003). Estrogens are known to improve synaptic plasticity, and behavior is affected in ovariectomized female rats (reviewed in Arevalo et al (2015)).

Consistent with the central role of PPARα in FA catabolism (Staels et al, 1998), PPARα null mice exhibit greater lipid accumulation (Chung et al, 2018). PPARα gene expression levels dose-dependently control liver metabolism, inflammation, and atherogenesis (Lalloyer et al, 2011) and are tightly regulated by cellular content in FA. Low plasma-free FA increases PPARα mRNA level in human skeletal muscle (Watt et al, 2004), whereas lipid accumulation decreases PPARα expression in the renal tubular epithelial region in rats (Chung et al, 2018), suggesting that the availability of FA is important for the regulation of *Ppara* gene transcription. Accumulation of FA has been previously reported in female but not male mice carrying FAD mutations (Barrier et al, 2010). Such FA accumulation could explain why PPARα expression is lower in female 5xFAD Tg mice.

Fibrates are PPARα agonists used in the treatment of hyper-triglyceridemia, mixed dyslipidemia, and also prevent the progression of atherosclerotic lesions (reviewed in Gross et al (2017)). Fenofibrate has been widely used, but its relatively low activity on PPARα led to the development of pemafibrate, a more potent and selective agonist for PPARα (Yamazaki et al, 2007; Hennuyer et al, 2016). In recent clinical studies, pemafibrate improved lipid profiles in patients with type 2 diabetes and hypertriglyceridemia (Araki et al, 2018) with a much higher efficacy than fenofibrate (Ishibashi et al, 2018). We show here that pemafibrate significantly improved hippocampal LTP in male but not in female Tg mice, confirming the involvement of PPARα in synaptic plasticity in a sex-specific manner.

Whereas hippocampal LTP was improved by both pemafibrate and bexarotene treatments of male Tg mice, pemafibrate administration increased expression of both GluN2A and GluA1 whereas bexarotene only increased expression of GluA1. Although PPAR/RXR heterodimers are permissive (Evans & Mangelsdorf, 2014), it was

previously demonstrated that the conformation of the ligand–receptor complexes and the nature of their interaction with co-regulators can differently modulate the transcription of target genes (Dowell et al, 1997; Schulman et al, 1998; Perez et al, 2012). We, therefore, hypothesize that because of their different affinities for different cofactors, bexarotene could up-regulate only GluA1, whereas pemafibrate is able to drive the expression of both GluA1 and GluN2A subunits. Consequently, LTP improvement observed upon RXR and PPARα activation relies mainly on GluA1, but we cannot exclude that GluN2A could also be involved when PPARα is activated by pemafibrate. Although this study strongly supports that targeting PPARα could be an effective strategy to improve synaptic plasticity deficits related to cognitive defects (D'Orio et al, 2018), it presents some limitations. Our study was limited to the 5xFAD mouse model of AD. Therefore, further investigations are needed to confirm whether PPARα could be an interesting target in other mouse models of neurodegenerative diseases including Alzheimer, Parkinson, and Huntington diseases, as well as multiple and amyotrophic lateral sclerosis, in which cognitive impairments occur. These mouse models do not fully recapitulate all pathological changes observed in patients and translating synaptic plasticity changes in mice with cognitive deficits in humans is challenging. However, based on the observations that bexarotene improves cognition in mouse models, we previously reported that Targretin (bexarotene) improved cognition in a patient with mild AD (Pierrot et al, 2016). In the same way, pemafibrate is used in human phase III clinical trials (Araki et al, 2018; Ishibashi et al, 2018), and investigating its effects on cognition in humans could be an interesting translational study based on our results.

Despite these limitations, we report here a sex-regulated gene dosage effect of PPARα on synaptic plasticity. In animal models, sex differences should be considered rather than making the choice of the best responder. In humans, sex differences exist in the vulnerability, incidence, manifestation, and treatment of numerous neurological and psychiatric diseases (Riecher-Rossler, 2017). Our results outline the importance to decipher sex differences in neurodegenerative diseases, including AD (Ferretti et al, 2018) with complex cognitive and neuropsychiatric symptoms, to define new sex-specific therapeutic strategies.

# Materials and Methods

### Animals

All animal procedures used in the study were carried out in accordance with the institutional and European guidelines as certified by the local Animal Ethics Committee. Both pregnant Wistar rats used for embryonic cell cultures of either sex were obtained from Université catholique de Louvain (UCL, Brussels, Belgium) animal facilities. All protocols were approved by the local ethical committee of the UCL. 5xFAD (Oakley et al, 2006) mice were obtained from Jackson Laboratories (strain: B6SJL-Tg (APPSwFlLon, PSEN1*M146L*L286V) 6799Vas/Mmjax), bred as heterozygous 5xFAD mice. 5–6-mo-old PPARα-deficient (*Ppara*$^{-/-}$) mice were used (Lee et al, 1995). Age-matched non-transgenic wild-type littermates were

used as controls. Experiments performed with male and female separately were indicated. Animals were housed on a 12-h light/dark cycle in standard animal care facilities.

## Reagents and antibodies

When unmentioned, reagents for cell culture, Western blotting, and calcium imaging were purchased from Thermo Fisher Scientific. Antibodies were purchased as indicated. Primary antibodies: mouse monoclonal anti-Glutamate Receptor 2 (6C4), rabbit monoclonal anti-GluA1 (C3T), and anti-GluN2A antibodies (Cat. No. MAB397, 04-855, and 07-632, respectively; Merck Millipore); mouse monoclonal anti-GluN2B (Cat. No. 610417; BD Biosciences); goat polyclonal anti-SYNGAP (Cat. No. LS-C154908; Bio-Connect Life Sciences); mouse monoclonal anti-ABCA1 (Cat. No. ab18180; Abcam); mouse monoclonal anti-$\alpha$ tubulin and mouse monoclonal anti-MAP2 (Cat. No. T6074 M4403 and A2066, respectively; Sigma-Aldrich); and anti-Transferrin Receptor mouse monoclonal antibody (H68.4) (Cat. No. 13-6800; Thermo Fisher Scientific). Secondary antibodies: donkey anti-rabbit and anti-mouse IgG horseradish (HRP) linked (Cat. No. NA934 and NA931, respectively; GE Healthcare-Life Sciences); Alexa Fluor 647 goat anti-mouse IgG1, 488 chicken anti-goat IgG (H+L), and 568 goat anti-rabbit IgG (H+L) (Cat. No. A21240, A21467, and A11036, respectively; Thermo Fisher Scientific).

## Cell cultures

Hippocampal and cortical neuronal cultures were prepared from embryonic day 17 (E17) to E18 Wistar rats or P0-P1 pups from $Ppara^{-/-}$ and wild-type ($Wt$) mice from the same genetic background of either sex. Pregnant rats and mice were euthanized with $CO_2$. Hippocampi and cortices were isolated as previously described (Seibenhener & Wooten, 2012; Pierrot et al, 2013) with slight modifications. Briefly, hippocampal neurons were dissociated by incubation (15 min, 37°C) in 0.25% Trypsin–EDTA and triturated in Hank's balanced salt solution without $CaCl_2$ and $MgCl_2$ supplemented with 10 mM Hepes. Hippocampal and cortical cells were plated in culture dishes (1.5 and $4 \times 10^5$ cells/cm$^2$, respectively) pretreated with 10 $\mu$g/ml poly-L-lysine (Sigma-Aldrich) in PBS and cultured for 13–14 d in vitro in Neurobasal medium supplemented with 2% (vol/vol) B-27 medium and 0.5 mM L-glutamine without antibiotic solution before analyses. Hippocampal cells were pre-plated in a neuronal plating medium (MEM with Earl's salt supplemented with 2 mM glutamine, 330 $\mu$M D-Glucose [Cat. No. G7528; Sigma-Aldrich], and 5% fetal bovine serum [Cat. No. S1820; Biowest]) during 4–5 h before Neurobasal medium described above. The cultures were maintained at 37°C under a 5% $CO_2$ atmosphere and half of the medium was renewed every 2–3 d.

## Recombinant viruses and infection

$Ppara$ and scramble shRNA containing AAV were purchased from Vectors Biolabs (Cat. No. shADV-269120 and 7045, respectively). For Ppara silencing, an AAV9-ShPpara ($3.9 \times 10^{13}$ GC/ml), containing an shRNA sequence (CCCTTATCTGAAGAATTCTTA) targeting both rat and mouse Ppara (Genbank RefSeq: NM_013196) and enhanced GFP (eGFP) reporter gene, was produced. The expressions of Ppara and

eGFP were driven by a U6 and a CMV promoter, respectively. An AAV9-GFP-U6-scramble shRNA (AAV-ShSc, $4.7 \times 10^{13}$ GC/ml) was used as a control. Cultures were transduced on fourth day in vitro (4 DIV) using AAV-ShPpara or AAV-ShSc at a multiplicity of infection of 12,000 overnight. Then, the infection medium was replaced by a fresh culture medium every 2 d up to analysis (between 13 and 14 DIV).

## Treatments and oral gavage

Treatments: cultured cells and hippocampal organotypic tissue cultures were treated for 24 h with 100 nM and 300 nM bexarotene in 0.0002% DMSO (Targretin), respectively. Control cells were treated with 0.0002% DMSO. For cell calcium imaging, neurons were challenged with 50 $\mu$M NMDA (Cat. No. M3262; Sigma-Aldrich) or 50 $\mu$M AMPA (Cat. No. 1074; Tocris) in the presence of 1 $\mu$M tetrodotoxin, a selective inhibitor of $Na^+$ channel conductance used to block spontaneous $[Ca^{2+}]_i$ transients in neurons (Cat. No. 1078; Tocris). For LTP measurements done on acute hippocampal slices from transgenic 5xFAD mice (9–10 mo) injected with AAV-Sh constructs, slices were treated with 4 $\mu$M bexarotene in artificial cerebrospinal fluid (aCSF) (see below) for 2h30 min before high-frequency stimulation (see below).

Oral gavage: age-matched non-transgenic wild-type and 5xFAD mice (9–10 mo) were treated for 12 d by oral gavage with 100 mg/kg/d b.wt. bexarotene or vehicle (water) or with 1 mg/kg/d b.wt. pemafibrate (at 12 mo) (Hennuyer et al, 2016) (Cat. No. HY-17618; MedChemExpress) or vehicle (water 0.1% Tween 80).

## Biotinylation and purification of plasma membrane–associated proteins

13–14 DIV–cultured cells seeded at $4 \times 10^5$ cells/cm$^2$ were washed with Krebs–Hepes buffer (see below). The cells were incubated with 1.6 ml of EZ-Link Sulfo-NHS-Biotin (Cat. No. 21217; Thermo Fisher Scientific) at 1.5 mg/ml in PBS for 30 min at 4°C with mild shaking. The cells were then washed twice with cold PBS containing 100 mM glycine and incubated with the same solution for 45 min at 4°C to quench the unbound biotin reagent. The cells were solubilized in lysis buffer containing 25 mM Tris–HCl, pH 6.8, 0.5% (vol/vol) Triton X-100, and 0.5% (vol/vol) Nonidet P-40 supplemented with proteases inhibitors for 1 h at 4°C with vigorous shaking. After centrifugation at 16,000 $g$ at 4°C for 20 min, 300 $\mu$l of supernatant were incubated with an equal volume of Pierce Streptavidin Agarose beads suspension (Cat. No. 20349; Thermo Fisher Scientific) for 1 h at room temperature. After centrifugation (16,000 $g$, 15 min, 4°C), supernatants were collected for analysis of the non-biotinylated intracellular fraction. Biotinylated cell surface proteins contained in the pellet were washed two times with 600 $\mu$l lysis buffer and two times in Krebs–Hepes buffer. The samples were eluted in 50 $\mu$l loading buffer (see below), boiled at 95°C for 5 min.

## Western blotting

Cells in culture were washed, scraped off in PBS, and centrifuged for 2 min at 16,000 $g$. Pellets were sonicated in lysis buffer (125 mM Tris [pH 6.8], 20% glycerol, and 4% sodium dodecyl sulfate) with

cOmplete Protease Inhibitor Cocktail (Cat. No. 11697498001; Roche). For brain protein extraction, the samples were homogenized in RIPA buffer (1% NP40, 0.5% deoxycholic acid, 0.1% SDS, 150 mM NaCl, 1 mM EDTA, and 50 mM Tris, pH 7.4) containing protease and phosphatase inhibitors cocktail (Cat. No. 04906837001; Roche). The samples were clarified by centrifugation at 20,000 $g$, and the protein concentration was determined using a Bicinchoninic Acid Assay (BCA) kit. The samples were heated for 10 min at 70°C in loading buffer (lysis buffer containing 10% 2-mercaptoethanol and 0.004% bromophenol blue).

Cell and brain lysates (40 and 60 $\mu$g of proteins, respectively) were analyzed by Western blotting using 4–12% Nupage bis-Tris gels. Nitrocellulose membranes were incubated overnight at 4°C with the following primary antibodies: anti-Glutamate Receptor 2 (GluA2, 1:1,000); anti-GluA1 (1:500); anti-GluN2A (1:250); anti-GluN2B (1:500); anti-ABCA1 (1:1,000); anti-$\alpha$ tubulin (1:4,000); and anti-Transf R (1:1,000). Blots were incubated with HRP peroxidase-conjugated secondary antibodies (1:10,000), revealed by ECL (Cat. No. ORT2655-2755; Amersham Pharmacia), and quantified using the Quantity One software (Bio-Rad Laboratories). $\alpha$-tubulin or Transf R was used as internal standards to normalize protein load in gels.

### RNA extraction and real-time PCR

Total RNA was isolated by TriPure Isolation Reagent (Cat. No. 11667165001; Roche) according to the manufacturer's protocol. RNA samples were resuspended in DEPC-treated water. Reverse transcription was carried out with the iScript cDNA synthesis kit (Cat. No. 1708891; Bio-Rad Laboratories) using 1 $\mu$g of total RNA in a total volume reaction of 20 $\mu$l. Real-time PCR was performed for the amplification of cDNAs with specific primers (Sigma-Aldrich, see Table S1).

Real-time PCR was carried out in a total volume of 25 $\mu$l containing 8 ng cDNA template, 0.3 $\mu$M of the appropriate primers, and the IQ SYBR Green Supermix 1× (Cat. No. 1708885; Bio-Rad Laboratories). The PCR protocol consisted of 40 amplification cycles (95°C for 30 s, 60°C for 45 s, and 79°C for 15 s) and was performed using an iCycler IQ Multicolor Real-Time PCR Detection System (Bio-Rad Laboratories), to determine the threshold cycle (Ct). Melting curves were performed to detect nonspecific amplification products. A standard curve was established for each target gene using fourfold serial dilutions (from 100 to 0.097 ng) of a cDNA template mix prepared in the same conditions. The differences between the Ct of one condition and the control were measured, and each sample was normalized with the relative expression levels of *Gapdh*.

### Cytosolic-free Ca²⁺ measurement in single neurons

For cytosolic-free $Ca^{2+}$ measurement, all recordings were carried out at 37°C in Krebs–Hepes buffer (10 mM Hepes, 135 mM NaCl, 6 mM KCl, 2 mM $CaCl_2$, 1.2 mM $MgCl_2$, and 10 mM glucose, pH 7.4), as previously described (Doshina et al, 2017). Briefly, 50 $\mu$M NMDA or AMPA were perfused with Krebs–Hepes buffer in the incubation chamber. Neurons were plated at a density of $1.8 \times 10^5$ cells/cm² on 15-mm round glass coverslips precoated with 10 $\mu$g/ml poly-L-lysine in PBS. 13–14 DIV–cultured cells were incubated in the dark in

the presence of the $Ca^{2+}$ indicator fura-2 acetoxymethylester (Fura-2 AM; Cat. No. F1225) at a final concentration of 2 $\mu$M in Krebs–Hepes buffer for 30 min at room temperature. Coverslips were then washed and mounted in a heated (37°C) microscope chamber (1 ml). The cells were alternately excited (1 or 2 Hz) at 340 and 380 nm for 100 ms using a Lambda DG-4 Ultra High Speed Wavelength Switcher (Sutter Instrument) coupled to a Zeiss Axiovert 200 M inverted microscope (X20 fluorescence objective) (Zeiss Belgium). Images were acquired using a Zeiss Axiocam camera coupled to a 510-nm emission filter and analyzed with Axiovision software. A total of 70–80 neurons were studied in each experiment, and non-neuronal cells were excluded from the analysis as previously described by Pickering and coworkers (Pickering et al, 2008). Changes in intracellular calcium fluorescence were estimated from fluorescence emission intensity ratio F340/F380 (ΔF) obtained after excitation of cells at wavelengths of 340 and 380 nm. These changes were expressed as normalized fluorescence where every measurement of ΔF was divided by the basal fluorescence (F0) value corresponding to the mean of signals measured during a period of 20 s in basal condition (before NMDA or AMPA). NMDA and AMPA responses were defined as a change of ΔF greater than 10% relative to F0.

### Stereotaxic injections

For stereotaxic surgery, 9–10-mo-old 5xFAD mice were anesthetized by intraperitoneal injection (i.p.) with a mixture of 160 mg/kg b.wt. ketamine (Nimatek; Eurovet Animal Health BV) and 20 mg/kg b.wt. xylazine (ROMPUN; Bayer). Ipsi and contralateral stereotaxic injections (left and right hemisphere, respectively) were performed at two sites in the hippocampal CA1 region (A/P, −1.8; L, ±1.1; D/V, −1.3 and A/P, −2.5; L, ±2.0; D/V, −1.5) millimeter relative to bregma (Paxinos & Franklin, 2001). Ipsilateral AAV-ShPpara ($3.9 \times 10^{13}$ GC/ml) or contralateral AAV-ShSc ($4.7 \times 10^{13}$ GC/ml) stereotaxic injection (5 $\mu$l in total of each; 2.5 $\mu$l per site of injections) were performed using a 10-$\mu$l Hamilton syringe (Filter Service, Cat. No. HA 7635-01) at a speed of 1 $\mu$l per min. After injection, the needle was kept in place for additional 3 min before gentle withdrawal. All analyses were performed 3 wk postinjection.

### Electrophysiology—LTP

Male and female PPAR$\alpha$-deficient ($Ppara^{-/-}$) and transgenic (Tg) 5xFAD mice at 5–6 and 9–10 mo of age, respectively, were anesthetized with pentobarbital (Nembutal, i.p. 100 mg/kg b.wt.) and decapitated. Age-matched wild-type (Wt) mice of the same genetic background were used as controls. The hippocampus was dissected and cut into 450-$\mu$m-thick slices with a tissue chopper. The slices were transferred into the recording chamber and kept in interface at 28°C for 1.5 h. Hippocampal slices were perfused with aCSF with the following composition: 124 mM NaCl, 5 mM KCl, 26 mM $NaHCO_3$, 1.24 mM $NaH_2PO_4$, 2.5 mM $CaCl_2$, 1.3 mM $MgSO_4$, and 10 mM glucose, bubbled with a mixture of 95% $O_2$ and 5% $CO_2$. The perfusion rate of aCSF was 1 ml/min. LTP was induced by applying one train (100 Hz, 1 s). A bipolar twisted nickel-chrome electrode (50 $\mu$m each) was used to stimulate Schaffer's collaterals. Extracellular field excitatory postsynaptic potentials (fEPSPs) were recorded in the stratum

radiatum of the CA1 region with low-resistance (2–5 MΩ) glass microelectrodes filled with aCSF (Villers & Ris, 2013). Test stimuli were biphasic (0.08 ms for each pulse) constant-voltage pulses delivered every minute with an intensity adjusted to evoke an approximate 40% maximal response. The slope of the fEPSP was measured on the average of four consecutive responses. Stimulation, data acquisition, and analysis were performed using the WinLTP program (Anderson & Collingridge, 2007) (www.winltp.com). For each slice, the fEPSP slopes were normalized with respect to the mean slope of the fEPSPs recorded during the 30-min period preceding induction of LTP.

### Confocal microscopy and image processing and analysis

Cells were seeded at $10^5$ cells/cm$^2$ on 15-mm round glass coverslips precoated with 10 µg/ml poly-L-lysine in PBS, fixed 15 min with 4% vol/vol formaldehyde at room temperature, then washed in PBS, and permeabilized 1 h with 0.4% Triton X-100 (vol/vol) in PBS containing 3% bovine serum albumin (Cat. No. A7906; Sigma-Aldrich). After three washes in PBS, the cells were incubated 1 h at room temperature with primary antibodies: anti-SynGAP (1:50), anti-MAP2 (1:1,000), and anti-GluA1 (1:100). After three PBS washes, the cells were incubated for 1 h with 5 µg/ml Alexa-labelled secondary antibodies (1:200). After three additional PBS washes, preparations were mounted in EverBrite (Cat. No. 23003; VWR) and were examined with an LSM 510 META confocal microscope (Zeiss) using a Plan-Apochromat 63×/1.4 oil DIC objective. The non-overlapping between GluA1 and SynGAP was determined on line intensity profiles. After threshold value determination to define the effective dynamic range, peaks were identified and classified into three categories: (i) only red, indicating non-overlapping of GluA1 with SynGAP; (ii) only green, indicating non-overlapping of SynGAP with GluA1; and (iii) red + green, indicating overlapping between GluA1 and SynGAP. The abundance of peaks in each category was then expressed as percentage of total peaks. Cluster size of GluA1 puncta was quantified using AxioVision 4.8.2. Images were first resampled to isolate the red channel for analysis (resampling step) and then segmented to isolate only the grey values between 80 and 255 (segmentation step). Images were then binarized, with the white areas corresponding to the GluA1 clusters (binary scrap step), and small holes in these clusters were filled up (binary fill step). After visual comparison of these white areas with the clusters on the initial pictures, the average area of clusters was measured with the AxioVision software in the "automatic measurement" mode, and data were exported in Excel for calculation and statistical analysis.

### Statistics

Statistical analyses were performed using GraphPad Prism 7.01 (GraphPad Software). The Shapiro–Wilk test was used to test for the normality of data. Parametric testing procedures (*t* test or one-way analysis of variance [ANOVA] followed by Bonferroni's multiple-comparison posttest when many subgroups were compared) were applied for normally distributed data, otherwise nonparametric tests were used (Mann–Whitney or Kruskal–Wallis tests followed by Dunn's multiple-comparison posttest when many subgroups were compared). Total number of samples (n) analyzed in all

experimental conditions (number of repeated measurements) is indicated in figures legends. Results were presented as mean ± SEM and statistical significance was set at $P$ values < 0.05 (two-tailed tests, except for Morris water maze experiments, only a one-sided $P$ value is presented) (*$P$ < 0.05, **$P$ < 0.01; ***$P$ < 0.001). For LTP, statistical differences of the means (±SEM) were measured on the last 30 min before the end of the recording. Graphical data are represented as plot data with individual points overlaid (Supplemental Data 1).

## Supplementary Information

## Acknowledgements

We thank the Fondation Louvain for support to N Pierrot; the Netherlands Brain Bank for providing us with human brain samples; F Saez-Orellana, PhD (IoNS, Brussels, Belgium), for discussion; and Pr. L Hue for his critical evaluation of the results and editing the manuscript. This work was supported by the Belgian Fonds pour la Recherche Scientifique, Interuniversity Attraction Poles Program-Belgian State-Belgian Science Policy, The Belgian Fonds de la Recherche Scientifique Médicale, the Queen Elisabeth Medical Foundation, and the Fondation pour la Recherche sur la Maladie d'Alzheimer.

### Author Contributions

N Pierrot: conceptualization, formal analysis, supervision, validation, investigation, methodology, and writing—original draft, review, and editing.
L Ris: formal analysis and methodology.
I-C Stancu: methodology.
A Doshina: methodology.
F Ribeiro: methodology.
D Tyteca: methodology and writing—original draft.
E Baugé: methodology.
F Lalloyer: methodology.
L Malong: methodology.
O Schakman: methodology.
K Leroy: methodology.
P Kienlen-Campard: project administration.
P Gailly: methodology and writing—original draft.
J-P Brion: methodology.
I Dewachter: methodology.
B Staels: conceptualization, methodology, and writing—original draft.
J-N Octave: conceptualization, formal analysis, supervision, funding acquisition, validation, investigation, and writing—original draft, review, and editing.

### Conflict of Interest Statement

The authors declare that they have no conflict of interest.

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
