## [Reviewer comments · Life Science Alliance]

Life Science Alliance

Sex regulated gene dosage effect of PPAR α on synaptic plasticity.

Nathalie Pierrot, Laurence Ris, Ilie-Cosmin STANCU, Anna DOSHINA, Floriane RIBEIRO, Donatienne Tyteca, Eric Baugé, Fanny LALLOYER, Liza MALONG, Olivier Schakman, Karelle LEROY, Pascal Kienlen-Campard, Philippe Gailly, Jean-Pierre Brion, Ilse DEWACHTER, Bart Staels, and Jean-Noel Octave

DOI: <https://doi.org/10.26508/lsa.201800262>

Corresponding author(s): Nathalie Pierrot, Université catholique de Louvain

Review Timeline:

Submission Date:	2018-11-30
Editorial Decision:	2019-01-03
Revision Received:	2019-02-14
Editorial Decision:	2019-02-28
Revision Received:	2019-03-01
Accepted:	2019-03-11

Scientific Editor: Andrea Leibfried

Transaction Report:

January 3, 2019

Re: Life Science Alliance manuscript #LSA-2018-00262-T

Nathalie Pierrot
Universite catholiqu de Louvain
Laboratoire de Pharmacologie experimentale
Avenue Hippocrate 54 FARL5410
Brussels
Belgium

Dear Dr. Pierrot,

Thank you for submitting your manuscript entitled "Sex regulated gene dosage effect of PPAR α on synaptic plasticity" to Life Science Alliance. The manuscript was assessed by expert reviewers, whose comments are appended to this letter.

As you will see, the reviewers appreciate your results. However, as pointed out by reviewer #2, important controls are lacking. We would thus like to invite you to provide a revised version of this work. Importantly, we expect that the requested controls get added, while the additional mouse work as proposed by the reviewer in point 2 is not mandatory for acceptance here.

Thank you for this interesting contribution to Life Science Alliance. We are looking forward to receiving your revised manuscript.

Sincerely,

- A letter addressing the reviewers' comments point by point.
- An editable version of the final text (.DOC or .DOCX) is needed for copyediting (no PDFs).
- High-resolution figure, supplementary figure and video files uploaded as individual files: See our detailed guidelines for preparing your production-ready images, <http://life-science-alliance.org/authorguide>
- Summary blurb (enter in submission system): A short text summarizing in a single sentence the study (max. 200 characters including spaces). This text is used in conjunction with the titles of papers, hence should be informative and complementary to the title and running title. It should describe the context and significance of the findings for a general readership; it should be written in the present tense and refer to the work in the third person. Author names should not be mentioned.

B. MANUSCRIPT ORGANIZATION AND FORMATTING:

Full guidelines are available on our Instructions for Authors page, <http://life-science-alliance.org/authorguide>

Reviewer #1 (Comments to the Authors (Required)):

Pierrot and colleagues submit the manuscript "Sex regulated gene dosage effect of PPARalpha on synaptic plasticity".

In a broad methodological set-up based on cell-culture, electrophysiology, Ca²⁺-imaging and

immunoblot analyses on brain homogenates from transgenic animals, the authors demonstrate that PPARalpha mediates the improvement of hippocampal synaptic plasticity upon RXR activation in an Alzheimer disease mouse model. RXR activation has been further shown to improve AMPA responses and GluA1 expression. Moreover, the authors provide experimental evidence that PPARalpha deficiency impairs LTP and GluA1 expression in male mice, and interestingly, that RXR-mediated improvement of synaptic plasticity is sex dependent.

The authors conclude that sex differences in hippocampal synaptic plasticity observed in mice are related to differences in PPARalpha expression levels and implicate further a sex regulated gene dosage effect of PPARalpha on synaptic plasticity.

The study is highly original and well performed. The experiments are statistically sound and all analyses are graphically well presented. The authors have used different experimental strategies to confirm their hypothesis. The manuscript is excellently written, the results are comprehensibly described. The authors have discussed also the limitation of their study and the Alzheimer mouse model and have proposed future experimental strategies.

Minor concerns:

The authors are encouraged to provide full-length blots (at least as supplemental data) of their immunoblot analyses and to also indicate specific and non-specific bands within these blots.

Reviewer #2 (Comments to the Authors (Required)):

In the present study, Pierrot et al., investigated the effect of nuclear receptors activation on hippocampal synaptic plasticity, using genetic and pharmacological manipulations, electrophysiology, imaging, biochemistry and real-time qPCR. First, they report that RXR activation using bexarotene improves LTP in a transgenic model with cognitive deficits. Using real-time qPCR, biochemistry and immunocytochemistry, they provide evidence that this effect results from a selective upregulation of GluA1 expression at excitatory synapses. Using PPARa null mutants, they show that the LTP improvement following RXR pharmacological activation requires the expression of PPARa. Second, they report that the effect mediated by PPARa is specific to males which express higher levels of PPARa and GluA1 compared to females. Furthermore, male but not female PPARa null mutants display impaired hippocampal LTP. Using pemafibrate, they provide evidence that selective activation of PPARa improve LTP in male cognition-impaired mice but not in females.

Overall, the experiments are sound and well performed and the question raised is indeed interesting. The results are logically and nicely introduced, presented and discussed. However, I have a few concerns that need to be addressed prior to publication :

1) I'm wondering to what extent the bexarotene and the pemafibrate are selective for RXR and PPARa activation, respectively. This is an important point to address or confirm as authors' main conclusions depend on it. In particular, the authors show that pemafibrate administration increases expression of both GluN2A and GluA1, leaving the possibility that LTP improvement not only relies on GluA1 but also on GluN2A. This is in contrast with bexarotene administration, which increases GluA1 only. In addition, PPARa null mutant or PPARa knock-down-selectively downregulate GluA1 levels but not GluN2A. These effects suggest that pemafibrate may have some offtarget/secondary effects, which need to be discussed.

2) To confirm the role of PPAR α , could the authors perform experiment where they overexpress or re-express PPAR α on WT, PPAR α -/-, or Tg background to see if they affect LTP similarly to the activation of PPAR α with pemafibrate?

3) There are some important missing controls: effect of bexarotene and pemafibrate on WT, male versus female, mice. In particular, I wonder whether administration of bexarotene or pemafibrate improves LTP also in WT mice (control missing in Fig1A), as it increases basal GluA1 in cultured neurons. In addition, if LTP difference between WT males and females relies on a differential expression of PPAR α , administration of these compounds in females should increase LTP to the same magnitude as in males. The experiment in Fig.4E, F is interesting but the control with no bexarotene is missing.

4) I found intriguing that bexarotene treatment is sufficient to fully recover LTP in Tg mice as GluN2A and GluN2B levels remain rather low (<50% of the control). In such condition, I would expect LTP to be still impaired at least to some extent compared to control + vehicle as partial blockade of NMDARs was shown to partially impair LTP (see for instance Jiang et al., Molecular Psychiatry, 2016).

Minor comments:

1) It seems that LTP magnitude is much more variable in females Ppara-/- than in female WT. Could it be that some female individuals display impaired LTP? The author should discuss / explain this variability.

2) I find puzzling the immunostaining of DIV 13-14 cultured neurons, as neurons look immature and aspiny for their age. Also, the authors should provide some evidence that their antibody is specific to GluA1.

3) In supplementary figures EV3, the authors should provide the quantification for PPAR α and CREB signal (immunostainings).

Reviewer #1 (Comments to the Authors (Required)):

Pierrot and colleagues submit the manuscript "Sex regulated gene dosage effect of PPARalpha on synaptic plasticity".

In a broad methodological set-up based on cell-culture, electrophysiology, Ca²⁺-imaging and immunoblot analyses on brain homogenates from transgenic animals, the authors demonstrate that PPARalpha mediates the improvement of hippocampal synaptic plasticity upon RXR activation in an Alzheimer disease mouse model. RXR activation has been further shown to improve AMPA responses and GluA1 expression. Moreover, the authors provide experimental evidence that PPARalpha deficiency impairs LTP and GluA1 expression in male mice, and interestingly, that RXR-mediated improvement of synaptic plasticity is sex dependent.

The authors conclude that sex differences in hippocampal synaptic plasticity observed in mice are related to differences in PPARalpha expression levels and implicate further a sex regulated gene dosage effect of PPARalpha on synaptic plasticity.

The study is highly original and well performed. The experiments are statistically sound and all analyses are graphically well presented. The authors have used different experimental strategies to confirm their hypothesis. The manuscript is excellently written, the results are comprehensibly described. The authors have discussed also the limitation of their study and the Alzheimer mouse model and have proposed future experimental strategies.

Minor concerns:

The authors are encouraged to provide full-length blots (at least as supplemental data) of their immunoblot analyses and to also indicate specific and non-specific bands within these blots.

Response:

As requested by the reviewer, we provide original full-length blots for the main and supplementary figures of the manuscript. Specific bands used for immunoblot analyses have been highlighted in blue. The original immunoblots are provided as PDF supplementary files entitled "Source Data".

Reviewer #2 (Comments to the Authors (Required)):

In the present study, Pierrot et al., investigated the effect of nuclear receptors activation on hippocampal synaptic plasticity, using genetic and pharmacological manipulations, electrophysiology, imaging, biochemistry and real-time qPCR. First, they report that RXR activation using bexarotene improves LTP in a transgenic model with cognitive deficits. Using real-time qPCR, biochemistry and immunocytochemistry, they provide evidence that this effect results from a selective upregulation of GluA1 expression at excitatory synapses. Using PPAR α null mutants, they show that the LTP improvement following RXR pharmacological activation requires the expression of PPAR α . Second, they report that the effect mediated by PPAR α is specific to males which express higher levels of PPAR α and GluA1 compared to females. Furthermore, male but not female PPAR α null mutants display impaired hippocampal LTP. Using pemafibrate, they provide evidence that selective activation of PPAR α improve LTP in male cognition-impaired mice but not in females.

Overall, the experiments are sound and well performed and the question raised is indeed interesting. The results are logically and nicely introduced, presented and discussed. However, I have a few concerns that need to be addressed prior to publication:

1) I'm wondering to what extent the bexarotene and the pemafibrate are selective for RXR and PPAR α activation, respectively. This is an important point to address or confirm as authors' main conclusions depend on it. In particular, the authors show that pemafibrate administration increases expression of both GluN2A and GluA1, leaving the possibility that LTP improvement not only relies on GluA1 but also on GluN2A. This is in contrast with bexarotene administration, which increases GluA1 only.

Response:

Bexarotene was previously described to be a selective ligand for RXR (Boehm et al, 1995) whereas pemafibrate is presented as a selective agonist for PPAR α (Hennuyer et al, 2016; Yamazaki et al, 2007) [as indicated on page 15 in the Discussion section of the manuscript].

We were indeed very surprised to see that pemafibrate administration increases expression of both GluN2A and GluA1, while bexarotene administration increases GluA1 only. Nevertheless, our results confirm those obtained by Roy et al. (Roy et al, 2013) showing that absence of PPAR α decreased the expression of both AMPA and NMDA receptors subunits in hippocampi of 6-8 weeks old Ppara $^{-/-}$ mice and in E18 cultured hippocampal Ppara $^{-/-}$ neurons.

Although PPAR/RXR heterodimers are permissive (Evans & Mangelsdorf, 2014) [as mentioned in the Introduction section of the manuscript (page 3)], it was previously

demonstrated that the conformation of the ligand-receptor complexes and the nature of their interaction with co-regulators is able to modulate differently the transcription of target genes (Dowell et al, 1997; Perez et al, 2012; Schulman et al, 1998). Therefore, we speculate that different affinities for different cofactors could explain why RXR activation by bexarotene upregulates only GluA1 whereas PPAR α activation by pemafibrate is able to drive the expression of both GluA1 and GluN2A subunits. Consequently, LTP improvement observed upon RXR activation relies mainly on GluA1, but we cannot exclude that GluN2A could also be involved when PPAR α is activated by pemafibrate.

As requested by the reviewer, this has been discussed in the new version of the manuscript (Discussion section page 16) as followed: "While hippocampal LTP was improved by both pemafibrate and bexarotene treatments of male Tg mice, pemafibrate administration increased expression of both GluN2A and GluA1 whereas bexarotene only increased expression of GluA1. Although PPAR/RXR heterodimers are permissive (Evans & Mangelsdorf, 2014), it was previously demonstrated that the conformation of the ligand-receptor complexes and the nature of their interaction with co-regulators can differently modulate the transcription of target genes (Dowell et al, 1997; Perez et al, 2012; Schulman et al, 1998). We therefore hypothesize that due to their different affinities for different cofactors, bexarotene could upregulate only GluA1 whereas pemafibrate is able to drive the expression of both GluA1 and GluN2A subunits. Consequently, LTP improvement observed upon RXR and PPAR α activation relies mainly on GluA1, but we cannot exclude that GluN2A could also be involved when PPAR α is activated by pemafibrate".

The corresponding references mentioned above have been added to the reference list.

"In addition, PPAR α null mutant or PPAR α knock-down-selectively downregulate GluA1 levels but not GluN2A. These effects suggest that pemafibrate may have some offtarget/secondary effects, which need to be discussed".

Response:

We agree with the reviewer that our data indicate that downregulation of PPAR α expression has a major effect on the downregulation of GluA1. Following PPAR α knock-down in male Tg mice, the decrease in PPAR α expression could not be strong enough to observe a decrease in the expression of GluN2A together with GluA1 in hippocampal lysates. However, we observed a decreased expression of GluN2A in Ppara^{-/-} cortical cells as compared to Wt, confirming previous data (Roy et al, 2013). Such a downregulation is illustrated in the Western blot presented here below.

GluN2A expression is decreased in Ppara^{-/-} cortical cells. Representative Western blots of cortical cells in culture from Wt and Ppara^{-/-} mice. At 13-14 DIV, expression of GluN2A was monitored. Blot was further probed using an α -tubulin antibody. GluN2A / α tubulin ratios were quantified and results were expressed as percentage (mean \pm s.e.m.) of Wt; ***P<0.001, Student's t test.

2) To confirm the role of PPAR α , could the authors perform experiment where they overexpress or re-express PPAR α on WT, PPAR α ^{-/-}, or Tg background to see if they affect LTP similarly to the activation of PPAR α with pemafibrate?

Response:

We agree with the reviewer that the re-expression of PPAR α in a PPAR α ^{-/-} Tg background would be a very elegant way to confirm the role of PPAR α . These experiments are under progress but could not be completed in the time allotted for resubmission of the manuscript.

3) There are some important missing controls: effect of bexarotene and pemafibrate on WT, male versus female, mice. In particular, I wonder whether administration of bexarotene or pemafibrate improves LTP also in WT mice (control missing in Fig1A), as it increases basal GluA1 in cultured neurons.

Response:

As requested by the reviewer, we measured LTP in Wt mice before and after treatment with bexarotene by gavage (100mg/kg/day) for 12 days. As illustrated in the figure here below, bexarotene does not significantly modify LTP, even if a slight improvement of LTP by bexarotene is observed between 120 and 180 minutes. Bexarotene is able to cross the blood-brain barrier to some extent (Landreth et al, 2013; Tesseur et al, 2013), but a breakdown of the blood-brain barrier was previously demonstrated in 5XFAD mice (reviewed in (Montagne et al, 2017)). This could explain why LTP improvement is observed in 5xFAD but not in Wt treated mice. Of course, the blood-brain barrier does not exist in cultured neurons in which bexarotene is able to increase basal GluA1 expression.

A new figure has been added as Supplementary Figure S1A (formerly Figure EV1). Accordingly, the following sentence was added on page 5 of the new version of the manuscript: "Impaired LTP found in Tg 5xFAD hippocampus was recovered (P<0.0001) after oral administration of bexarotene for 12 days and became similar to vehicle treated control mice (Fig 1A). Bexarotene did not improve LTP of Wt mice (Fig S1A). The efficiency of the treatment of Tg mice could result from a breakdown of the blood-brain barrier in 5XFAD mice (Montagne et al. 3151 2017)". This recovery of LTP in 5xFAD mice was observed together with improved cognition in the object recognition and spatial navigation tasks, which was independent of amyloid plaque load in different regions of the brain (Figs S1B-1E)".

Figure S1A

The corresponding figure's legend and the numbering of figures has been adapted as followed [page 38 in the new version of the manuscript]:

Figure S1. RXR activation improves cognition in an AD mouse model, independently of brain amyloid plaque load.

(A) 9-10 months old wild-type (Wt) mice treated with bexarotene (bex) (100mg/kg/day) or vehicle (veh) by gavage (12 days). CA1 LTP in hippocampal slices of Wt + bex (n=4) was compared to Wt + veh (n=4). $P > 0.05$, Student's t test. **(B)** 9-10 months old 5xFAD transgenic (Tg) and wild type (Wt) mice treated with bexarotene (bex) (100mg/kg/day) or vehicle (veh) by gavage. ...”

“In addition, if LTP difference between WT males and females relies on a differential expression of PPAR α , administration of these compounds in females should increase LTP to the same magnitude as in males.

Response:

As mentioned above, we were not able to show any effect of the treatments on LTP in WT animals. Our results indicate a correlation between PPAR α expression and GluA1 expression in WT males and females (Fig. 3 A-D), and GluA1 subunit plays a critical role in synaptic plasticity. However, many different pathways, which do not involve PPAR α , can control GluA1 expression, as mentioned in the discussion of the revised version of our manuscript (page 14). Therefore, we do not conclude that LTP difference between WT males and females relies only on a differential expression of PPAR α . Nevertheless, a two times higher expression of PPAR α in males than in females induces a PPAR α -mediated regulation of GluA1 expression only in males. This suggests that a threshold level of PPAR α expression is needed to regulate GluA1 expression, and this level is not reached in females. Consequently, Tg females are insensitive to the bexarotene (Fig 4F) or the pemafibrate treatment (Fig. 5B).

The discussion has been modified accordingly (page 14): “Many different pathways, which do not involve PPAR α , can control GluA1 expression, and we cannot conclude that LTP difference between WT males and females relies only on a differential expression of PPAR α .

Nevertheless, a two times higher expression of PPAR α in males than in females induces a PPAR α -mediated regulation of GluA1 expression only in males. This suggests that a threshold level of PPAR α expression is needed to regulate GluA1 expression, and this level is not reached in females, which are insensitive to bexarotene treatment.”

“The experiment in Fig.4E, F is interesting but the control with no bexarotene is missing”.

Response:

As requested by the reviewer, LTP was measured in hippocampal slices from male Tg mice with or without bexarotene, and the results are presented in the figure here below. This new figure has been added in the new version of the manuscript as Supplementary Figure S7C (formerly Figure EV7). Accordingly, the following sentence was added on page 11 of the new version of the manuscript: “We first measured LTP on hippocampal slices from male Tg mice treated or not for 2.5 h with bexarotene 4 μ M (Fig S7C). The results were similar to those obtained following gavage of Tg mice”. LTP was then measured on hippocampal slices from male and female Tg mice...”

The corresponding figure’s legend has been adapted as followed [page 44 of the new version of the manuscript]:

Figure S7. PPAR α and GluA1 expression in male and female Tg mice.

...(C) CA1 LTP in hippocampal slices from male (σ) transgenic (Tg) 5xFAD mice (9-10 mo) perfused with 4 μ M bexarotene (bex, n=4) or with vehicle (veh, n=2), *** P<0.001, Student’s t test. Data information: data are presented as mean \pm SEM.

4) I found intriguing that bexarotene treatment is sufficient to fully recover LTP in Tg mice as GluN2A and GluN2B levels remain rather low (<50% of the control). In such condition, I would expect LTP to be still impaired at least to some extent compared to control + vehicle as partial blockade of NMDARs was shown to partially impair LTP (see for instance Jiang et al., Molecular Psychiatry, 2016).

We were indeed surprised to observe that RXR activation with bexarotene was sufficient to recover LTP in Tg mice while levels of GluN2A and GluN2B were decreased by 50% compared to wild-type mice. This highlights the major effect of GluA1 upregulation on LTP recovery. In their paper, Jiang et al. (Molecular Psychiatry, 2016) report that partial “inhibition of NMDARs with APV (2 μM) impairs, but does not block, LTP in wild-type mice”, indicating that decrease in NMDA-mediated synaptic transmission impairs LTP. In their study, LTP was induced by a standard induction protocol (2 trains of 100 Hz for 1 s separated by 20 s) and analyzed during 40 min on hippocampal slices from young wild-type mice, while we used one train of stimulation (100 Hz, 1s) and analyzed LTP after 180 min in old Tg mice. These could explain differences observed.

Minor comments:

- 1) It seems that LTP magnitude is much more variable in females Ppara^{-/-} than in female WT. Could it be that some female individuals display impaired LTP? The author should discuss / explain this variability.

Response:

In general, we have observed more variability in LTP measured in females. This could result from hormonal fluctuations occurring across the female estrus cycle. Indeed, it was shown that ovarian steroids, such as 17-β estradiol, modulate hippocampal dendritic spine density, synaptic activity and synaptic plasticity across the estrus cycle, with the highest level of excitatory effect during proestrus (Good et al, 1999; Scharfman et al, 2003; Woolley & McEwen, 1992). In addition, PPARs participate in steroidogenesis, cytokine production and angiogenesis during estrous cycle (Froment et al, 2006; Matsuda et al, 2013). Therefore, absence of PPARα in female mice could further increase the variability in LTP measured in females.

- 2) I find puzzling the immunostaining of DIV 13-14 cultured neurons, as neurons look immature and aspiny for their age.

Response:

We measured the maturity of neuronal networks in culture by calcium imaging. We previously demonstrated that calcium oscillations are present at DIV 13-14 (Doshina et al, 2017).

Also, the authors should provide some evidence that their antibody is specific to GluA1.

Response:

Although this antibody was previously utilized to detect GluA1 in synaptosomal membranes from rat dorsomedial striatum homogenates (Wang et al, 2012), the specificity of the anti-GluA1 antibody used in this study was tested as requested by the reviewer. We performed a Western blotting on cell lysates from CHO and cortical cells as presented below. GluA1 expression is only detect in neuronal lysates and not in CHO cell lines.

Specific GluA1 expression in cortical cells in culture. Western blot of cell lysates (20 μ g) from CHO and cortical cells in culture. GluA1 was detected with the GluA1 04-855 antibody from Merck-Millipore. Blot was further probe with the anti-actin antibody.

3) In supplementary figures EV3, the authors should provide the quantification for PPAR α and CREB signal (immunostainings).

As suggested by the reviewer, quantification for PPAR α and CREB signals have been performed in the Figure S3 (formerly Figure EV3) in the new version of the manuscript.

The corresponding figure's legend has been adapted as followed [page 41 in the new version of the manuscript:

Figure S3. RXR activation increases CREB and PPAR α labelling in neurons and astrocytes. "(B, C) ... CREB and PPAR α signals were quantified in Co and bex treated cells. Data are normalized to Co (CREB: n=7 and PPAR α : n=4 of each, analysed in 3 independent

experiments, * $P < 0.05$, Student's *t*-test). Data information: data are presented as mean \pm SEM."

Bibliography

Boehm MF, Zhang L, Zhi L, McClurg MR, Berger E, Wagoner M, Mais DE, Suto CM, Davies JA, Heyman RA et al (1995) Design and synthesis of potent retinoid X receptor selective ligands that induce apoptosis in leukemia cells. *J Med Chem* 38: 3146-3155

Doshina A, Gourgue F, Onizuka M, Opsomer R, Wang P, Ando K, Tasiaux B, Dewachter I, Kienlen-Campard P, Brion JP et al (2017) Cortical cells reveal APP as a new player in the regulation of GABAergic neurotransmission. *Sci Rep* 7: 370

Dowell P, Ishmael JE, Avram D, Peterson VJ, Nevriy DJ, Leid M (1997) p300 functions as a coactivator for the peroxisome proliferator-activated receptor alpha. *J Biol Chem* 272: 33435-33443

Evans RM, Mangelsdorf DJ (2014) Nuclear Receptors, RXR, and the Big Bang. *Cell* 157: 255-266

Froment P, Gizard F, Defever D, Staels B, Dupont J, Monget P (2006) Peroxisome proliferator-activated receptors in reproductive tissues: from gametogenesis to parturition. *J Endocrinol* 189: 199-209

Good M, Day M, Muir JL (1999) Cyclical changes in endogenous levels of oestrogen modulate the induction of LTD and LTP in the hippocampal CA1 region. *Eur J Neurosci* 11: 4476-4480

Hennuyer N, Duplan I, Paquet C, Vanhoutte J, Woitrain E, Touche V, Colin S, Vallez E, Lestavel S, Lefebvre P et al (2016) The novel selective PPARalpha modulator (SPPARMalpha) pemafibrate improves dyslipidemia, enhances reverse cholesterol transport and decreases inflammation and atherosclerosis. *Atherosclerosis* 249: 200-208

Landreth GE, Cramer PE, Lakner MM, Cirrito JR, Wesson DW, Brunden KR, Wilson DA (2013) Response to comments on "ApoE-directed therapeutics rapidly clear beta-amyloid and reverse deficits in AD mouse models". *Science* 340: 924-992g

Matsuda S, Kobayashi M, Kitagishi Y (2013) Expression and Function of PPARs in Placenta. *PPAR Res* 2013: 256508

Montagne A, Zhao Z, Zlokovic BV (2017) Alzheimer's disease: A matter of blood-brain barrier dysfunction? *J Exp Med* 214: 3151-3169

Perez E, Bourguet W, Gronemeyer H, de Lera AR (2012) Modulation of RXR function through ligand design. *Biochim Biophys Acta* 1821: 57-69

Roy A, Jana M, Corbett GT, Ramaswamy S, Kordower JH, Gonzalez FJ, Pahan K (2013) Regulation of cyclic AMP response element binding and hippocampal plasticity-related genes by peroxisome proliferator-activated receptor alpha. *Cell Rep* 4: 724-737

Scharfman HE, Mercurio TC, Goodman JH, Wilson MA, MacLusky NJ (2003) Hippocampal excitability increases during the estrous cycle in the rat: a potential role for brain-derived neurotrophic factor. *J Neurosci* 23: 11641-11652

Schulman IG, Shao G, Heyman RA (1998) Transactivation by retinoid X receptor-peroxisome proliferator-activated receptor gamma (PPARgamma) heterodimers: intermolecular synergy requires only the PPARgamma hormone-dependent activation function. *Mol Cell Biol* 18: 3483-3494

Tesseur I, Lo AC, Roberfroid A, Dietvorst S, Van BB, Borgers M, Gijzen H, Moechars D, Mercken M, Kemp J et al (2013) Comment on "ApoE-directed therapeutics rapidly clear beta-amyloid and reverse deficits in AD mouse models". *Science* 340: 924-992e

Wang J, Ben Hamida S, Darcq E, Zhu W, Gibb SL, Lanfranco MF, Carnicella S, Ron D (2012) Ethanol-mediated facilitation of AMPA receptor function in the dorsomedial striatum: implications for alcohol drinking behavior. *J Neurosci* 32: 15124-15132

Woolley CS, McEwen BS (1992) Estradiol mediates fluctuation in hippocampal synapse density during the estrous cycle in the adult rat. *J Neurosci* 12: 2549-2554

Yamazaki Y, Abe K, Toma T, Nishikawa M, Ozawa H, Okuda A, Araki T, Oda S, Inoue K, Shibuya K et al (2007) Design and synthesis of highly potent and selective human peroxisome proliferator-activated receptor alpha agonists. *Bioorg Med Chem Lett* 17: 4689-4693

February 28, 2019

RE: Life Science Alliance Manuscript #LSA-2018-00262-TR

Dr. Nathalie Pierrot
Université catholique de Louvain
Institute of Neuroscience
Avenue Mounier 53, SSS/IONS/CEMO-Bte B1.53.03
Brussels 1200
Belgium

Dear Dr. Pierrot,

Thank you for submitting your revised manuscript entitled "Sex regulated gene dosage effect of PPAR α on synaptic plasticity". Reviewer #2 evaluated the revised version and appreciates the introduced changes. We would thus be happy to publish your paper in Life Science Alliance pending final revisions necessary to meet our formatting guidelines:

- please incorporate the supplementary method information into the main manuscript; we do not have space limitations and doing so will help the readers
- please link your profile in our submission system to your ORCID iD, you should have received an email with instructions on how to do so
- we follow ICMJE author contribution guidelines, please check these for all contributing authors (<http://www.icmje.org/recommendations/browse/roles-and-responsibilities/defining-the-role-of-authors-and-contributors.html>)
- we appreciate the uploaded source data and would like to ask you to indicate the lane spliced out in Fig1E (total fraction-GluA1); to mark in the source data the a-tub lanes used in Fig5F, third blot; to clarify which lanes are depicted in SFig6B for ABCA1

A. FINAL FILES:

-- High-resolution figure, supplementary figure and video files uploaded as individual files: See our

detailed guidelines for preparing your production-ready images, <http://www.life-science-alliance.org/authors>

B. MANUSCRIPT ORGANIZATION AND FORMATTING:

Sincerely,

Andrea Leibfried, PhD
Executive Editor
Life Science Alliance
Meyershofstr. 1
69117 Heidelberg, Germany
t +49 6221 8891 502
e a.leibfried@life-science-alliance.org

Reviewer #2 (Comments to the Authors (Required)):

The authors have addressed all the main concerns I had. They provide new convincing and strong evidence to support their conclusions. I therefore recommend the publication of this study in Life Science Alliance.

March 11, 2019

RE: Life Science Alliance Manuscript #LSA-2018-00262-TRR

Dr. Nathalie Pierrot
Université catholique de Louvain
Institute of Neuroscience
Avenue Mounier 53, SSS/IONS/CEMO-Bte B1.53.03
Brussels 1200
Belgium

Dear Dr. Pierrot,

Thank you for submitting your Research Article entitled "Sex regulated gene dosage effect of PPAR α on synaptic plasticity.". It is a pleasure to let you know that your manuscript is now accepted for publication in Life Science Alliance. Congratulations on this interesting work.

DISTRIBUTION OF MATERIALS:

Again, congratulations on a very nice paper. I hope you found the review process to be constructive and are pleased with how the manuscript was handled editorially. We look forward to future exciting submissions from your lab.

Sincerely,
